# Geometric Control of Out-of-Distribution Shift in Safe Offline RL

**Zhiqi Zhuang** [1]  **Di Wu** [1][2]  **Benoit Boulet** [1]

## Abstract

Safe offline reinforcement learning (RL) requires optimizing policies within the support of static datasets while satisfying strict safety constraints. Although recent latent generative policies achieve strong empirical performance, they rely heavily on implicit regularization and lack systematic control over distributional shift during policy improvement. In this work, we propose a geometric control framework that leverages the bijective structure of conditional normalizing flows to provide a tractable mechanism to regulate distributional deviation of the policy. By constraining divergence in the latent base space, we derive tractable upper bounds on the induced Wasserstein distance and total variation of the policy distribution, establishing an analyzable connection between latent geometry and downstream behaviors. This insight motivates a decoupled architecture: a flow prior shapes a feasibility-weighted latent manifold using Hamilton–Jacobi reachability signals, while a latent refiner performs geometrically constrained optimization directly in the base space. Across multiple safe RL benchmarks, our method achieves consistently low violation rates with competitive returns, highlighting the benefits of structured geometric regularization.

## 1. Introduction

Safe offline reinforcement learning (RL) aims to learn policies that maximize return while satisfying stringent safety constraints using only static datasets, without online interaction (Levine et al., 2020). This setting poses a fundamental dual challenge: policies must improve beyond the behavior policy to achieve higher returns, yet strictly adhere to the

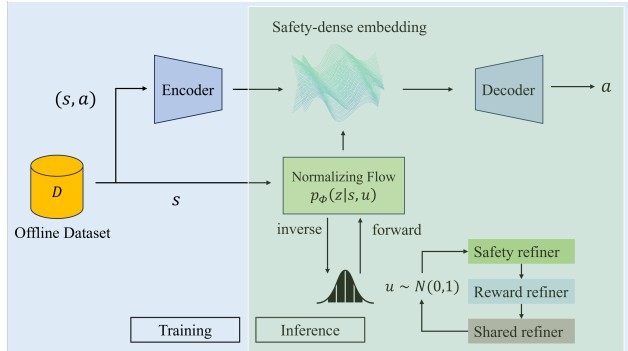

*Figure 1.* Overview of the proposed method. An encoder maps $(s, a)$ into a safety-dense latent embedding. A conditional normalizing flow prior $p_\phi(z|s)$ induced by $u \sim \mathcal{N}(0, I)$ serves as the prior, providing exact forward/inverse transforms between the base and latent spaces; a decoder then reconstructs actions $a$ from $z$. At inference, three refiners (safety, reward, and a shared refiner) operate in the *base* Gaussian space to adjust samples toward high-density, in-support regions—maximizing return while suppressing OOD actions and enforcing safety constraints.

underlying data support to avoid out-of-distribution (OOD) actions that can trigger catastrophic safety violations (Fujimoto et al., 2019; Kumar et al., 2020). While training from logs enables deployment in safety-critical domains such as robotics (Kalashnikov et al., 2018; Wu et al., 2024), autonomous driving (Kiran et al., 2021; Zhang et al., 2025), and industrial control (Yu et al., 2025; Wang et al., 2025), the tension between *policy improvement* and *distributional shift* remains a central bottleneck (Kushwaha et al., 2025; Wu et al., 2024).

State-of-the-art approaches increasingly rely on latent generative models to capture complex, multi-modal behavior distributions. Prominent examples include conditional variational autoencoders (CVAEs) (Zhou et al., 2021; Fujimoto et al., 2019; Koirala et al., 2024) and, more recently, diffusion models (Wang et al., 2022; Hansen-Estruch et al., 2023; Zheng et al., 2024). Although empirically effective, these methods typically impose OOD regularization *implicitly*—through reconstruction objectives, weighted sampling, or support constraints (Kostrikov et al., 2021; Chen et al., 2022). Moreover, dominant paradigms such as diffusion models lack tractable likelihoods, making it mathematically difficult to explicitly quantify and regulate how policy up-

[1]Department of Electrical and Computer Engineering, McGill University, Montreal, QC, Canada [2]Department of Mechanical, Industrial and Aerospace Engineering, Concordia University, Montreal, QC, Canada. Correspondence to: Di Wu <di.wu5@mcgill.ca>.

*Proceedings of the 43rd International Conference on Machine Learning*, Seoul, South Korea. PMLR 306, 2026. Copyright 2026 by the author(s).

*Table 1.* Representative generative latent(-space) policy methods for offline (safe) RL.

| Method | Backbone | Safety-aware? | Likelihood | OOD control |
|---|---|---|---|---|
| PLAS (Zhou et al., 2021) | CVAE | No | Approx. | Implicit (latent manifold) |
| LSPC (Koirala et al., 2024) | CVAE | Yes | Approx. | Implicit (bounded latent) |
| LDGC (Venkatraman et al., 2023) | Diffusion | No | Implicit | Implicit (batch-constrained) |
| FISOR (Zheng et al., 2024) | Diffusion | Yes | Implicit | Implicit (HJ-weighted data) |
| CNF (Akimov et al., 2022) | Flow | No | Exact | Implicit (bounded latent) |
| **GSCO (ours)** | **Flow** | **Yes** | **Exact** | **Explicit (base-space reg)** |

dates deviate from the learned data manifold. As a result, safety guarantees often rely on heuristic tuning or conservative penalties rather than principled, controllable mechanisms (Xu et al., 2022; Liu et al., 2023b).

In this work, we argue that *principled geometric control* can be critical for principled and analyzable regulation of policy deviation. We leverage the unique properties of conditional normalizing flows (NFs) (Algren et al., 2023; Zhai et al., 2024)—namely, their bijectivity and exact likelihood evaluation—to establish a direct and tractable connection between latent-space divergence and downstream policy shift. Specifically, by constraining divergence in the latent base space, we derive upper bounds on the induced Wasserstein distance and total variation of the policy distribution (Lemma 4.2 and Corollary 4.3), transforming the challenge of OOD control into a measurable and controllable constraint on latent geometry.

Building on this insight, we propose **GSCO** (Geometric Safe COntrol), a framework that decouples *safety-aware manifold modeling* from *policy optimization*. A flow-based prior learns a feasibility-weighted latent manifold shaped by Hamilton–Jacobi (HJ) reachability signals (Bansal et al., 2017; Fisac et al., 2018), concentrating density on empirically safe regions. Policy improvement is then performed through geometrically constrained refinement in the latent base space using decoupled objectives for reward optimization, safety shaping, and deviation regularization. Importantly, the invertible structure of the representation maintains consistency with the deviation analysis throughout refinement, enabling stable multi-objective optimization under purely offline supervision.

Our main contributions are:

- We formulate safe offline RL as a geometric control problem in a bijective latent space, and derive tractable bounds linking base-space divergence to action-space distributional shift.

- We propose a decoupled architecture that integrates feasibility-shaped flow representations with structured latent refinement to jointly address reward optimization, safety enforcement, and OOD control.

- We demonstrate strong empirical performance on stan-

dard safe offline benchmarks, achieving consistently low violation rates with competitive returns.

## 2. Related Works

Offline safe RL aims to learn constraint-satisfying policies from fixed datasets without risky online interaction. Early methods incorporate penalty or Lagrangian terms into value learning, such as CPQ (Xu et al., 2022), BCQ-Lag (Fujimoto et al., 2019), and BEAR-Lag (Liu et al., 2023a). Other approaches adopt distribution-correction techniques, e.g., COptiDICE (Lee et al., 2022), which model constrained stationary distributions. Sequence models such as CDT (Liu et al., 2023b) and SaFormer (Zhang et al., 2023b) encode safety via cost-aware conditioning in Decision Transformer frameworks. These methods typically enforce soft constraints and may allow occasional violations.

More recent work (Yu et al., 2022; Ganai et al., 2024; Lu et al., 2025) leverages Hamilton–Jacobi reachability to provide stricter state-wise safety guarantees. In parallel, another line of research learns generative policies or latent manifolds to encourage safe behavior. For example, LSPC (Koirala et al., 2024) learns a cost-sensitive latent policy using a CVAE prior, while FISOR (Zheng et al., 2024) combines diffusion-based behavior modeling with HJ-based feasibility guidance. While these methods achieve strong empirical safety, their generalization to OOD states is typically handled implicitly through the expressivity of the generative model or support constraints, and remains challenging in practice (Zhou et al., 2021; Akimov et al., 2022; Chen et al., 2022). Table 1 summarizes representative generative approaches, with further discussions in Appendix A.1.

Our method unifies these directions by combining a flow-based latent policy with explicit base-space geometric regularization, and by using HJ reachability not as a hard filter but as a feasibility signal to shape the latent geometry, yielding provable upper bounds on policy deviation in total variation.

## 3. Preliminaries

**Safe Offline Reinforcement Learning.** Safe RL is formulated as a constrained Markov decision process (CMDP)

$\mathcal{M} = (\mathcal{S}, \mathcal{A}, T, r, h, \gamma)$, where $h(s) \leq 0$ indicates safety. In the offline setting, learning is performed from a fixed dataset $\mathcal{D}$ collected by a behavior policy $\pi_\beta$. The objective is to maximize return while controlling safety violations and policy-induced distributional shift:

$$\max_\pi \ \mathbb{E}[V_r^\pi] \quad \text{s.t.} \quad \mathbb{E}[V_h^\pi] \leq \ell, \ \ D_{\mathrm{KL}}(\pi \| \pi_\beta) \leq \varepsilon. \quad (1)$$

where $V_r^\pi(s) = \mathbb{E}_\pi\big[\sum_{t=0}^\infty \gamma^t r(s_t, a_t) \mid s_0 = s\big]$ is the reward value function and $V_h^\pi(s) = \mathbb{E}_\pi\big[\sum_{t=0}^\infty \gamma^t \mathbb{I}(h(s_t) > 0) \mid s_0 = s\big]$ denotes the expected cumulative violation indicator. The KL constraint acts as a surrogate measure of distributional shift relative to the data policy. We focus on the strict zero-violation regime ($\ell = 0$), which motivates explicit geometric control of policy deviation.

**Normalizing Flows.** Normalizing flows (Kobyzev et al., 2020) define bijective mappings between a base distribution $u \sim \mathcal{N}(0, I)$ and a latent variable $z = f_\phi(u; \mathrm{cond})$. The log-density is given by the change-of-variables formula (derivation is deferred to Appendix B.1)

$$\log p_\phi(z \mid \mathrm{cond}) = \log p(u) + \log\left|\det \frac{\partial u}{\partial z}\right|, \quad (2)$$

enabling exact likelihood evaluation and tractable inverse mappings. We exploit this **bijective geometry** to propagate base-space divergence to action-space distributional shift with explicit bounds.

**Feasibility-Based Safety Signal.** We estimate a feasibility value following Hamilton–Jacobi reachability (Bansal et al., 2017; Yu et al., 2022). Let $h(s)$ denote the signed constraint. The feasibility value measures the worst-case violation along a future trajectory:

$$V_h(s) = \min_\pi \mathbb{E}_\pi[\max_{t \geq 0} h(s_t) \mid s_0 = s]. \quad (3)$$

In practice, $F(s)$ is approximated using a symbolic Bellman-style backup $V_h(s) \approx \max(h(s), \gamma \mathbb{E}[V_h(s')])$. We use $V_h(s)$ solely as an auxiliary geometric shaping signal; details are deferred to Appendix B.2.

# 4. Methodology

Building on the geometric perspective introduced in Section 1, we formulate safe offline RL as *geometrically constrained optimization* within a learned latent manifold. Instead of directly regulating policy deviation in the high-dimensional, multi-modal action space, we leverage the bijective structure of conditional normalizing flows to define a structured base space where distributional shift admits explicit and tractable control.

As illustrated in Figure 1, the framework decomposes into two stages. First, a flow-based prior learns a latent manifold

that captures the support of empirically feasible behaviors through feasibility-aware density shaping (Sec. 4.1). Second, policy improvement is performed by optimizing trajectories directly in the frozen base space, guided by decoupled safety and reward objectives and bounded by the derived geometric deviation guarantees (Sec. 4.2, Appendix B). This decoupling enables stable policy improvement while suppressing out-of-distribution drift.

## 4.1. Geometric Density Modeling via Bijective Flows

To enable principled control of out-of-distribution shift, we construct a structured latent manifold that captures the support of empirically feasible behaviors. Rather than directly regulating policy updates in the high-dimensional action space, we employ a conditional normalizing flow to define a bijective mapping between the action manifold $\mathcal{A}$ and a tractable base space $\mathcal{U} = \mathbb{R}^d$ equipped with a standard Gaussian measure $\mu = \mathcal{N}(0, I)$. This bijection allows distributional variation in the base space to be propagated to action-space deviation with explicit bounds, forming the geometric foundation of our approach.

**Manifold Support Estimation via Weighted ELBO.** The model consists of a flow prior $p_\phi(z|s)$, a posterior encoder $q_\psi(z|s, a)$, and a deterministic decoder $\pi_\theta(a|s, z)$. To bias the learned manifold toward feasible regions of the data distribution, we optimize a feasibility-weighted variational objective:

$$\begin{aligned}
\mathcal{L}_{\mathrm{ELBO}} = \ & \mathbb{E}_\mathcal{D} \mathbb{E}_{q_\psi}[-w(s, a) \log \pi_\theta(a|s, z)] \\
& + \beta \, \mathbb{E}_\mathcal{D}[w(s, a) \, D_{\mathrm{KL}}(q_\psi \| p_\phi)].
\end{aligned} \quad (4)$$

where $w(s, a) \in [0, 1]$ is a feasibility score derived from the auxiliary safety critic. The following lemma gives a precise variational interpretation of this objective in the special case $\beta = 1$.

**Lemma 4.1** (Feasibility-Weighted Variational Modeling). *Let $\tilde{p}_\mathcal{D}(s, a) = w(s, a) p_\mathcal{D}(s, a) / Z_w$, where $Z_w = \mathbb{E}_{p_\mathcal{D}}[w(s, a)]$, and define*

$$\begin{aligned}
P_\psi(s, a, z) &= \tilde{p}_\mathcal{D}(s, a) q_\psi(z|s, a), \\
Q_{\phi,\theta}(s, a, z) &= \tilde{p}_\mathcal{D}(s) p_\phi(z|s) \pi_\theta(a|s, z).
\end{aligned}$$

*For fixed weights and $\beta = 1$,*

$$Z_w^{-1} \mathcal{L}_{\mathrm{ELBO}} = D_{\mathrm{KL}}(P_\psi \| Q_{\phi,\theta}) - \mathbb{E}_{\tilde{p}_\mathcal{D}}[\log \tilde{p}_\mathcal{D}(a|s)]. \quad (5)$$

*Thus, minimizing $\mathcal{L}_{\mathrm{ELBO}}$ is equivalent, up to a model-independent constant, to minimizing a valid matched-joint KL divergence. For $\beta \neq 1$, Eq. (4) is used as a feasibility-weighted variational loss with scaled posterior-prior regularization, not as an exact KL projection. The proof is provided in Appendix B.3.*

**Base-Space Density Shaping.** While the weighted ELBO encourages the flow to capture the feasible support, it does not control the geometry of the base-space density. We therefore introduce an additional regularization term that aligns high-value feasible actions with high-density regions of the base distribution:

$$\mathcal{L}_{\text{shape}} = \mathbb{E}_{\mathcal{D}}\left[ \mathbf{I}_{\text{feas}}(s,a) \cdot e^{\frac{A_r(s,a)}{\beta_r}} \cdot \left\| f_\phi^{-1}(z|s) \right\|_2^2 \right], \quad (6)$$

where $f_\phi^{-1}$ denotes the exact inverse flow mapping from latent code to the base variable. This objective encourages feasible high-reward regions to concentrate near the high-density modes of the base distribution, yielding a smoother and better-conditioned optimization landscape for subsequent refinement.

**Full Objective.** We summarize the flow module's objective as:

$$\mathcal{L}_{\text{flow}} = \mathcal{L}_{\text{ELBO}} + \mathcal{L}_{\text{shape}} + \lambda_H \left( H_0 - \mathbb{E}_{q_\psi}[-\log q_\psi(z \mid s,a)] \right)_+, \quad (7)$$

where the final term softly enforces a minimum posterior entropy to prevent mode collapse.

**Frozen Decoder and Deviation Decomposition.** We consider the setting where the decoder $\pi_\theta$ is fixed, so that the geometric mapping induced by the flow and decoder remains invariant during downstream optimization. Under this assumption, policy deviation admits the following decomposition.

**Lemma 4.2** (Deviation Decomposition). *Let $\Pi_\theta(\cdot|s)$ denote the refined policy distribution and $\pi_0(\cdot|s)$ the pushforward of $\mathcal{N}(0,I)$ through the frozen flow and decoder. Assume a bounded density ratio between $\pi_0$ and the behavior policy on the data support. Then for any state $s$,*

$$D_{\text{KL}}\big(\Pi_\theta(\cdot|s) \,\|\, \pi_\beta(\cdot|s)\big) \leq \\ D_{\text{KL}}\big(q_u(\cdot|s) \,\|\, \mathcal{N}(0,I)\big) + C(s). \quad (8)$$

*where $C(s) = \log R_\theta(s)$ under a mild bounded density-ratio assumption between $\pi_0$ and $\pi_\beta$. The proof follows from the data-processing inequality and bijectivity (Appendix B.4).*

Lemma 4.2 shows that policy deviation from the data distribution can be regulated by directly controlling the divergence of the refined base distribution. This converts deviation control in the action space into a tractable constraint in the latent base space, which we exploit in the refinement stage.

**4.2. Geometrically Constrained Base-Space Refinement**

Given the feasibility-shaped latent manifold constructed in Sec. 4.1, we perform policy improvement by optimizing

directly in the base space $\mathcal{U}$. Unlike action-space perturbations, base-space updates preserve the manifold geometry induced by the bijective flow and the frozen decoder, enabling principled control of distributional shift.

**Geometric Motivation & Deviation Bounds.** Since the flow is invertible and the decoder is fixed, any change in the base distribution $q_u$ induces a deterministic and bounded shift in the resulting policy distribution. This connection is formalized in the following corollary, which motivates explicitly regulating base-space divergence to control OOD risk.

**Corollary 4.3** (A Sufficient Bound on Induced Distribution Shift). *Let $\pi = T_{s\#} f_{\phi\#} q_u$ be the refined policy and $\pi_0 = T_{s\#} f_{\phi\#} \mathcal{N}$ be the base policy, where $\mathcal{N}$ is the standard Gaussian. Let $L_g$ denote a (local) Lipschitz constant of the composite mapping on the learned manifold. Then for any state $s$ (proof in Appendix B.5),*

$$W_2(\pi, \pi_0) \leq L_g \sqrt{2\, D_{\text{KL}}(q_u \,\|\, \mathcal{N})}, \quad (9)$$

$$\text{TV}(\pi, \pi_\beta) \leq \sqrt{\tfrac{1}{2} D_{\text{KL}}(q_u \,\|\, \mathcal{N})} + \text{TV}(\pi_0, \pi_\beta). \quad (10)$$

Thus, $D_{\text{KL}}(q_u \| \mathcal{N})$ identifies the theoretical control target; in practice, we use the surrogate in Eq. (13) to encourage small base-space shift.

**Decomposed Refinement Principle.** Policy improvement must simultaneously satisfy three competing objectives: (i) increasing reward, (ii) maintaining feasibility, and (iii) remaining within the high-density manifold region to avoid out-of-distribution behavior. Optimizing these objectives with a single monolithic update is unstable and often leads to conflicting gradients. We therefore decompose refinement into three **semantic latent fields**, each specializing in one aspect of the objective: a reward field, a safety field, and a geometric regularization field.

**Sequential Latent Updates.** Starting from $u_0 \sim \mathcal{N}(0,I)$, we perform $T$ refinement steps in the base space. At each step, safety correction is applied first, followed by reward improvement and a lightweight invertible geometric projection:

$$u_t^h = u_t + f_h(s, u_t),$$
$$u_t^r = u_t^h + f_r(s, u_t^h),$$
$$u_{t+1} = g_\eta(u_t^T | s).$$

Here $f_h$ corrects unsafe deviations, $f_r$ promotes reward improvement, and the shared field $g_\eta$ regularizes the refined sample toward the base geometry.

Let $\bar{a}(s, u_T)$ denote the decoded mean action and reuse the critics $(Q_r, V_r)$ and $(Q_h, V_h)$. Each field is trained using a dedicated objective.

*(i) Reward field.*

$$\mathcal{L}_r = -\mathbb{E}_{\mathcal{D}}\big[w_r(s,a)\,\|\bar{a}(s,u_T) - a\|_2^2\big], \qquad (11)$$

where $w_r(s,a) = \exp\big([Q_r(s,a) - V_r(s)]/\beta_r\big) \cdot \mathbf{I}_{\text{feas}}$ emphasizes high-advantage actions while restricting updates to feasible regions.

*(ii) Safety field.*

$$\mathcal{L}_h = \mathbb{E}_{\mathcal{D}}\big[\phi(Q_h(s,\bar{a}) - V_h(s)) + w_h(s,a)\,\|\bar{a} - a\|_2^2\big], \ (12)$$

where $w_h(s,a) = \exp\big(-[Q_h(s,\bar{a}) - V_h(s)]/\beta_h\big) \cdot \mathbf{I}_{\text{feas}}$, and $\phi(\cdot)$ is a soft penalty. The first term suppresses the predicted violations, while the second anchors refinement to empirically safe data.

*(iii) Shared geometric field.* To explicitly align refinement with the base-space divergence in Corollary 4.3, we parameterize the shared field as a *single conditional affine coupling layer* $u_T = g_\eta(u_T' \,|\, s)$, where $u_T'$ denotes the output after reward and safety updates. This extremely lightweight invertible map enables exact evaluation of the local volume change via the log-determinant of the Jacobian.

We regularize the shared field by penalizing the induced base-space density shift:

$$\mathcal{L}_{\text{reg}} = \tfrac{1}{2}\|u_T\|_2^2 - \log|\det \nabla g_\eta(u_T' \,|\, s)| + \lambda_{\text{prox}}\|u_T - u_T'\|_2^2. \tag{13}$$

The Gaussian energy and log-determinant terms penalize large-norm base samples and local volume expansion induced by the invertible projection, while the proximal term limits distortion of the reward- and safety-guided proposal. Importantly, $L_{\text{reg}}$ is a tractable surrogate for the base-space divergence appearing in Corollary 4.3, rather than an exact estimator of $D_{\text{KL}}(q_u\|\mathcal{N})$. Directly estimating the density of the refined distribution $q_u$ is costly and unstable in high dimensions. The surrogate is therefore designed to encourage moderate base-space shift in a lightweight way that is geometrically aligned with the deviation analysis, while remaining fully decoupled from the reward and safety objectives.

**Refiner Objectives and Guarantees.** The three losses define complementary refinement objectives: reward improvement, safety correction, and geometric regularization. For compact notation, we summarize their weighted combination as

$$\mathcal{L}_{\text{ref}} = \lambda_r \mathcal{L}_r + \lambda_h \mathcal{L}_h + \lambda_{\text{reg}} \mathcal{L}_{\text{reg}}. \tag{14}$$

This formulation converts constrained policy improvement into a regularized base-space optimization problem and provides a principled characterization of policy deviation. Appendix B.6 derives upper bounds on the reward and cost policy gaps under standard regularity assumptions, quantifying the sensitivity of downstream performance and safety to base-space regularization.

## 4.3. Practical Implementation

We employ expectile regression to obtain in-sample, asymmetric value estimates that bias learning toward high-value actions without querying out-of-distribution actions, following standard in-support value estimation practice in offline RL (Kostrikov et al., 2021). Specifically, $V_r$ is trained via asymmetric expectile regression and $Q_r$ is bootstrapped toward $V_r$:

$$\mathcal{L}_{V_r} = \mathbb{E}_{(s,a)\sim\mathcal{D}}\big[\rho_{\tau_r}\big(Q_r(s,a) - V_r(s)\big)\big], \qquad (15)$$

$$\mathcal{L}_{Q_r} = \mathbb{E}_{(s,a,s')\sim\mathcal{D}}\Big[\big(Q_r(s,a) - \hat{Q}_r(s,a)\big)^2\Big], \qquad (16)$$

where $\rho_\tau(u) = |\tau - \mathbf{1}\{u < 0\}|\,u^2$ denotes the expectile loss and $\hat{Q}_r(s,a) = r(s,a) + \gamma V_r(s')$ is the TD target. As summarized in Appendix D and Alg. 1, training proceeds in two stages. First, we jointly train the reward and safety critics together with the flow prior and decoder using offline transitions and the objectives in Sec. 4.1, establishing a feasibility-shaped latent manifold. Second, we freeze this base model and train the latent refiners in the base space using the objectives in Sec. 4.2. All components are trained purely offline from the fixed dataset. At inference time, we sample $u \sim \mathcal{N}(0, I)$, apply the expert refiner for $T$ steps to obtain $u_T$, and decode through the frozen flow and decoder to obtain the final action. Additional implementation details are provided in Appendix D.

## 5. Experiments

**Experiment Setup.** We evaluate the proposed method against several strong offline safe RL baselines across three widely-used benchmark environments: **Safety-Gymnasium** (Ji et al., 2023), **Bullet-Safety-Gym** (Gronauer, 2022) and **Safe Metadrive** (Li et al., 2022) from the DSRL suite (Liu et al., 2023a). We adopt *normalized return* and *normalized cost* as evaluation metrics, which we refer to as "reward" and "cost" for clarity and brevity. We set a uniform cost limit of 10 for all tasks.

**Baselines.** We compare our approach against five representative baselines: **(1) BCQL** (Fujimoto et al., 2019): A batch-constrained Q-learning with an adaptive Lagrangian penalty on constraint violations. **(2) CPQ** (Xu et al., 2022): A Q-learning method that penalizes unsafe and out-of-distribution state–action pairs. **(3) CDT** (Liu et al., 2023b): A transformer-based offline safe RL method that learns cost-conditioned action generators for constraint enforcement. **(4) LSPC** (Koirala et al., 2024): A latent safety-constrained approach that uses a conditional variational autoencoder to model safety in the latent space. **(5) FISOR** (Zheng et al., 2024): A feasibility-guided method that uses a diffusion model for policy sampling.

*Table 2.* Performance Comparison on DSRL benchmark. ↑ means the higher the better, ↓ means the lower the better. *Note:* **Bold**: safe policy; Gray: unsafe policy; **Bold blue**: best safe policy; **Bold**: second best safe policy

| Task | BCQL | | CPQ | | CDT | | FISOR | | LSPC | | GSCO (ours) | |
|---|---|---|---|---|---|---|---|---|---|---|---|---|
| | reward ↑ | cost ↓ | reward ↑ | cost ↓ | reward ↑ | cost ↓ | reward ↑ | cost ↓ | reward ↑ | cost ↓ | reward ↑ | cost ↓ |
| **Safety-Gymnasium** | | | | | | | | | | | | |
| CarButton1 | 0.16 | 4.20 | 0.13 | 2.44 | 0.21 | 1.60 | **-0.04** | **0.58** | -0.15 | 0.58 | **0.03** | **0.36** |
| CarButton2 | 0.07 | 3.47 | 0.17 | 7.05 | 0.13 | 1.58 | **-0.01** | **0.22** | -0.03 | 0.59 | **0.04** | **0.38** |
| CarPush1 | **0.09** | **0.56** | -0.14 | 0.80 | **0.31** | **0.40** | 0.26 | 1.23 | **0.21** | **0.13** | 0.20 | 0.04 |
| CarPush2 | **0.06** | **0.61** | 0.10 | 5.66 | 0.19 | 1.30 | **0.16** | **0.71** | 0.04 | 1.37 | **0.24** | **0.36** |
| CarGoal1 | **0.13** | **0.90** | 0.22 | 0.79 | 0.66 | 1.21 | **0.42** | **0.88** | 0.23 | 0.71 | **0.27** | **0.00** |
| CarGoal2 | 0.13 | 2.38 | 0.17 | 3.10 | 0.48 | 1.25 | 0.06 | 0.06 | **0.11** | **0.50** | **0.20** | **0.28** |
| AntVel | 0.29 | 2.08 | **-0.31** | **0.00** | **0.98** | **0.39** | 0.90 | 0.00 | **0.91** | **0.02** | 0.69 | 0.00 |
| HalfCheetahVel | 1.04 | 7.06 | 0.08 | 2.56 | **0.97** | **0.55** | 0.88 | 0.00 | 0.86 | 0.18 | **0.94** | **0.16** |
| SwimmerVel | 0.29 | 4.10 | 0.31 | 2.66 | 0.67 | 1.47 | **0.01** | **0.01** | 0.47 | 1.26 | **0.06** | **0.00** |
| **Safety-Gym Avg** | 0.25 | 2.82 | 0.08 | 2.78 | 0.51 | 1.08 | **0.29** | **0.40** | 0.29 | 0.59 | **0.33** | **0.18** |
| **Bullet-Safety-Gym** | | | | | | | | | | | | |
| AntRun | 0.05 | 4.63 | **0.13** | **0.01** | 0.69 | 1.24 | **0.45** | **0.76** | 0.94 | 1.46 | **0.52** | **0.00** |
| BallRun | **0.35** | **0.20** | 0.85 | 13.67 | **0.88** | **0.86** | 0.14 | 0.00 | 0.08 | 0.00 | **0.16** | **0.00** |
| CarRun | 0.75 | 2.51 | **0.75** | **0.52** | 0.99 | 1.47 | **0.80** | **0.00** | 0.75 | 0.22 | **0.87** | **0.00** |
| DroneRun | **0.65** | **0.71** | 0.26 | 0.44 | **0.71** | **0.60** | 0.41 | 0.57 | 0.62 | 1.34 | **0.59** | **0.02** |
| AntCircle | 0.61 | 1.42 | **0.00** | **0.00** | 0.46 | 2.74 | 0.23 | 0.00 | **0.40** | **0.78** | **0.45** | **0.25** |
| BallCircle | 0.79 | 1.20 | 0.40 | 4.37 | 0.79 | 1.64 | **0.45** | **0.00** | 0.29 | 1.83 | **0.46** | **0.00** |
| CarCircle | 0.64 | 1.80 | 0.49 | 4.48 | 0.70 | 1.20 | **0.34** | **0.00** | 0.28 | 0.04 | **0.66** | **0.06** |
| DroneCircle | 0.68 | 1.19 | -0.27 | 1.29 | 0.59 | 1.56 | **0.60** | **0.00** | 0.66 | 1.37 | **0.54** | **0.00** |
| **Bullet-SG Avg** | 0.57 | 1.71 | 0.33 | 3.10 | 0.73 | 1.41 | **0.43** | **0.17** | **0.50** | **0.88** | **0.54** | **0.04** |
| **Safe MetaDrive** | | | | | | | | | | | | |
| Easysparse | 0.94 | 9.25 | **-0.05** | **0.15** | 0.25 | 0.15 | **0.41** | **0.50** | 0.74 | 1.55 | **0.32** | **0.20** |
| Easymean | 0.99 | 7.22 | **-0.06** | **0.00** | 0.42 | 0.25 | **0.43** | **0.67** | **0.70** | **0.68** | 0.25 | 0.10 |
| Easydense | 0.20 | 1.76 | **-0.06** | **0.16** | 0.35 | 1.17 | 0.52 | 1.26 | 0.74 | 1.48 | **0.33** | **0.11** |
| Mediumsparse | 0.94 | 2.83 | -0.08 | 0.12 | 0.78 | 1.24 | **0.43** | **0.08** | **0.97** | **0.79** | 0.31 | 0.06 |
| Mediummean | 0.70 | 4.45 | -0.07 | 0.16 | 0.72 | 2.74 | 0.36 | 0.02 | **0.92** | **0.89** | **0.52** | **0.63** |
| Mediumdense | 0.76 | 3.90 | -0.08 | 0.10 | 0.70 | 2.62 | **0.51** | **0.39** | **0.87** | **0.88** | 0.33 | 0.07 |
| Hardsparse | 0.49 | 3.16 | **-0.05** | **0.10** | 0.26 | 0.46 | **0.33** | **0.24** | 0.52 | 1.32 | **0.35** | **0.34** |
| Hardmean | 0.29 | 3.80 | **-0.05** | **0.15** | 0.20 | 0.61 | 0.27 | 0.01 | **0.41** | **0.57** | **0.28** | **0.10** |
| Harddense | 0.42 | 2.95 | **-0.04** | **0.12** | 0.22 | 1.38 | **0.30** | **0.26** | 0.53 | 1.63 | **0.36** | **0.11** |
| **MetaDrive Avg** | 0.64 | 4.37 | **-0.06** | **0.12** | 0.45 | 1.18 | **0.40** | **0.38** | 0.71 | 1.09 | **0.34** | **0.19** |

**Main Results** Table 2 summarizes results on Safety-Gymnasium, Bullet-Safety-Gym, and Safe MetaDrive. Overall, our method learns safe policies with competitive returns. BCQL uses a Lagrangian trade-off but often fails to meet safety constraints; CPQ is more conservative and improves safety at the cost of reward; and CDT, though capable of high returns via target conditioning, tends to violate safety more frequently. FISOR and LSPC are strong baselines with distinct characteristics. FISOR produces uniformly safe but slightly conservative policies via feasibility guidance, while LSPC is more aggressive—seeking the most rewarding action in a learned safe latent space—which may exhibit reduced robustness under OOD states/actions. Our GSCO trains safety and shared refiners to concentrate probability mass in high-density regions of the encoder's latent space, which encourages actions to remain in better-supported regions and empirically correlates with improved safety. GSCO performs strongly on Safety-Gymnasium and Bullet-Safety-Gym, and is mildly conservative on Safe MetaDrive due to limited overlap between high-reward and low-cost regions, which complicates hard-constrained optimization. Even so, it enforces safety effectively, achieving consistently lower violation rates than the second-best method (e.g., 0.18 vs. 0.40 in Safety-Gymnasium, 0.04 vs. 0.88 in Bullet-Safety-Gym, and 0.19 vs. 0.38 in Safe MetaDrive) while maintaining strong performance.

## 6. Ablation Study and Analysis

**Justification of Each Refiner.** A core challenge in safe RL is reconciling reward maximization with safety constraints, which often induce competing optimization direc-

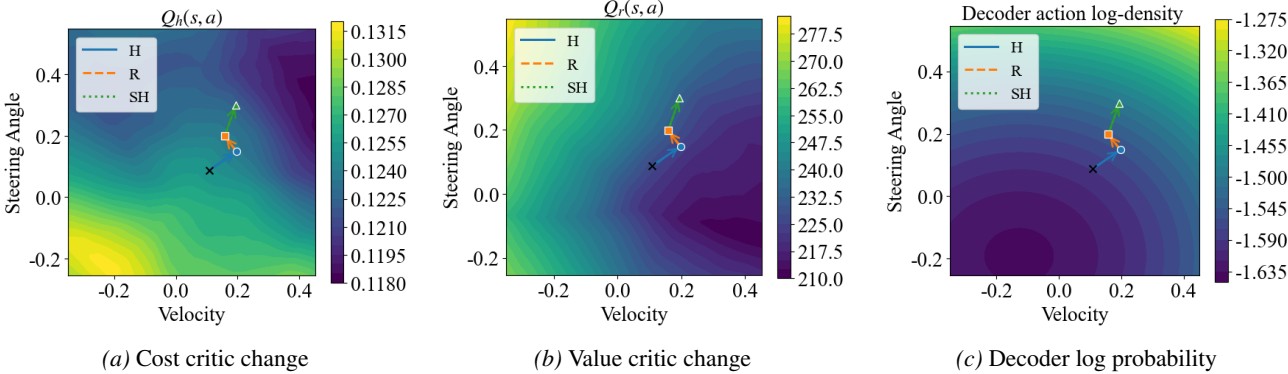

*Figure 2.* Example visualization of the refiner principle on `CarRun`. Each panel shows the 2D action space (velocity on the horizontal axis, steering angle on the vertical), where background colors indicate (a) feasibility (darker is safer), (b) reward (lighter is higher), and (c) decoder log-density (lighter is higher). The black cross is the base action from the flow prior, and colored curves (H, R, SH) show the refinement trajectories in base space, with triangles marking the final refined actions toward safer, higher-return, and data-supported regions.

*Table 3.* Ablations on HJ reachability.

| Task | w/o HJ | | GSCO | |
|---|---|---|---|---|
| | $r \uparrow$ | $c \downarrow$ | $r \uparrow$ | $c \downarrow$ |
| AntRun | 0.65 | 0.13 | 0.52 | 0.00 |
| BallRun | 0.08 | 0.14 | 0.16 | 0.00 |
| CarRun | 0.83 | 0.13 | 0.87 | 0.00 |
| DroneRun | 0.16 | 5.24 | 0.59 | 0.02 |
| AntCircle | 0.23 | 0.01 | 0.45 | 0.25 |
| BallCircle | 0.44 | 0.00 | 0.46 | 0.00 |
| CarCircle | 0.63 | 0.49 | 0.66 | 0.06 |
| DroneCircle | 0.56 | 0.67 | 0.54 | 0.00 |

tions. Figure 2 illustrates this tension on a fixed state from the `CarRun` task. Each panel visualizes the 2D action space (velocity vs. steering), where the background color encodes (a) feasibility $Q_h(s, a)$ (darker is safer), (b) reward value $Q_r(s, a)$ (lighter is higher return), and (c) decoder log-density $\log \pi_\theta(a \mid s)$ (lighter indicates higher density). In this state, high-reward and high-safety regions are largely disjoint and misaligned with the decoder's high-density region. As a result, the reward and safety refiners may produce conflicting updates that push the latent action toward low-density regions, increasing OOD risk and destabilizing refinement. The shared refiner mitigates this issue by explicitly regularizing refinement in the base space, effectively constraining trajectories to remain near the high-density manifold. This corresponds to regularizing the base-space energy, which is aligned with the theoretical control target in our geometric analysis. In practice, it acts as a stabilizing coordinator, damping overly aggressive updates and mediating conflicts between reward, safety, and data density objectives. Together, the three refiners form a complementary decomposition: reward drives improvement, safety enforces feasibility, and the shared refiner ensures geometric consistency and stable in-distribution optimization.

**HJ-feasibility Function.** We first assess the benefit of in-

corporating HJ reachability by replacing the feasibility function with a cost value function. states/actions whose cost falls below the empirical 75th percentile of zero-violation samples are treated as feasible and used for flow training, while refiner training is unchanged; we denote this variant as *w/o HJ*. As reported in Table 3, this heuristic thresholding yields noisier feasibility estimates, which in turn leads to higher evaluation costs and lower returns than the HJ-based approach. In contrast, HJ reachability propagates safety constraints through the dynamics, which is robust to sampling noise and uneven cost distributions. The results indicate that structured HJ reachability is crucial for stable constraint satisfaction in offline settings.

**The Order of Refinement.** We compare four refiner schedules on four tasks (`BallCircle`, `CarRun`, `AntCircle`, `DroneCircle`) to assess how sensitive GSCO is to the refiner order: two fixed orders (H→R→SH and R→H→SH), a random permutation, and a "No refine" baseline that samples directly from the flow prior. The results in Figure 3 show that all refiner variants substantially improve normalized return over No refine, confirming the benefit of latent refinement. Across all tasks, H→R→SH and R→H→SH achieve clearly higher return than no refinement baseline with low normalized cost, while the random-order variant is intermediate but with larger variability. We also observe a consistent trade-off pattern: H→R→SH generally yields lower cost with a strong but slightly lower return, whereas R→H→SH attains the highest return at the price of higher cost. This supports our design choice of using a fixed schedule with the shared refiner applied last so that it can consistently regularize and coordinate the preceding safety and reward updates.

**Reversed Expectile for Feasibility Function.** The reversed expectile parameter $\tau_h$ controls the conservativeness

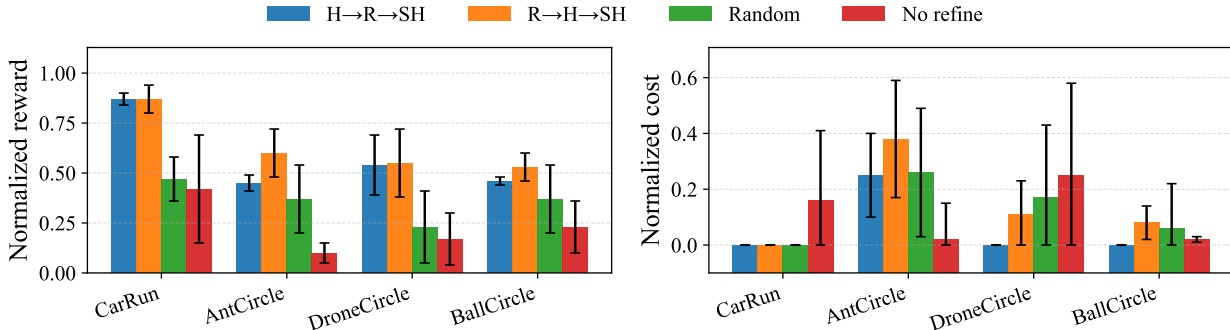

*Figure 3.* Effect of refiner order on normalized reward (left) and cost (right) across four tasks. Each group of bars corresponds to four refinement schedules (H→R→SH, R→H→SH, Random, No refine), with error bars showing one standard deviation.

*Table 4.* Ablations on the prior used.

| Task | Gaussian Prior | | Flow Prior | |
|---|---|---|---|---|
| | $r \uparrow$ | $c \downarrow$ | $r \uparrow$ | $c \downarrow$ |
| CarButton1 | -0.14 | 0.22 | 0.03 | 0.36 |
| CarButton2 | 0.01 | 0.82 | 0.04 | 0.38 |
| CarPush1 | 0.07 | 0.08 | 0.20 | 0.04 |
| CarPush2 | 0.06 | 0.00 | 0.24 | 0.36 |
| CarGoal1 | 0.06 | 0.00 | 0.27 | 0.00 |
| CarGoal2 | 0.05 | 0.74 | 0.20 | 0.28 |

*Table 5.* Sensitivity of the HJ-based feasibility classifier to the expectile parameter $\tau_h$. Recall/precision are computed on the offline buffer by treating steps from zero-cost trajectories as ground-truth safe.

| Task | Metric | $\tau_h$ | | | | |
|---|---|---|---|---|---|---|
| | | 0.6 | 0.7 | 0.8 | 0.9 | 0.95 |
| CarRun | Recall | 0.32 | 0.39 | 0.54 | 0.76 | 0.85 |
| | Precision | 0.76 | 0.68 | 0.51 | 0.24 | 0.21 |
| AntCircle | Recall | 0.04 | 0.08 | 0.27 | 0.79 | 0.88 |
| | Precision | 0.78 | 0.42 | 0.06 | 0.05 | 0.05 |

of the feasibility critic and therefore shapes the learned safe region. We sweep different values of $\tau_h$ to quantify the discrepancy between the HJ-based feasibility estimates and the empirical safe set extracted from the offline dataset. A smaller $\tau_h$ places more weight on lower $Q_h$ values, yielding a more pessimistic estimate of $V_h$ and consequently shrinking the induced feasible set $\{s \mid V_h(s) \leq 0\}$. This bias is expected to favour high precision but low recall with respect to the true safe set. Conversely, a larger $\tau_h$ produces a more optimistic critic, expanding the feasible region and trading precision for recall. Table 5 confirms this trend on both `CarRun` and `AntCircle`: recall consistently increases with $\tau_h$, while precision decreases. The trade-off is significantly more pronounced on `AntCircle`, whose safety boundary is more complex and harder to approximate from offline data. This suggests that tasks with more fragmented or nonlinear safe regions require a moderately optimistic critic (larger $\tau_h$) to achieve adequate coverage of feasible states, whereas simpler tasks can tolerate more conservative settings.

**Other Ablations.** We further examine the effect of the prior. As a comparison, we train a variant that replaces our flow-based prior with a Gaussian prior and report results in Table 4. The flow prior yields consistently higher returns while maintaining low costs, suggesting that the conditional invertible mapping provides a more structured and state-dependent latent geometry than an isotropic Gaussian prior. This supports our design choice of using a normal-

izing flow not only for likelihood evaluation, but also for shaping a base space in which safety-aware refinement can be performed more reliably. We also study the number of refinement steps $T$ at inference on `CarCircle`. We fix the refinement order: the safety expert is applied first, and the shared expert last. This design reflects our latent geometry—density concentrates on safety rather than reward—so early safety refinement places trajectories in high-density feasible regions before reward-oriented improvement. Intermediate refiners alternate between safety and reward experts, while the final shared refiner regularizes the resulting proposal toward the learned base geometry. As shown in Figure 4, increasing $T$ reduces cost and variability: larger $T$ allows more gradual latent-space correction and lowers measured OOD tendency, but an overly large $T$ can induce slightly more conservative behavior. In practice, an intermediate value, such as $T = 3$, provides a favorable reward–safety trade-off.

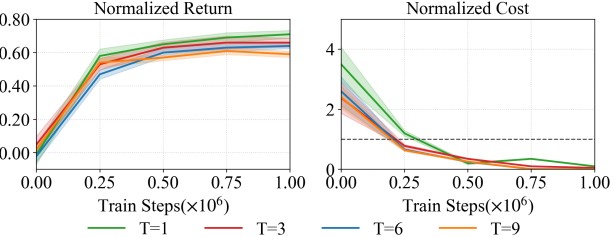

*Figure 4.* Ablation on the number of refinement steps.

*Table 6.* Direct OOD measurements on representative tasks. We report the state-conditioned OOD rate and OOD distance, both computed with respect to the same local feasible support extracted from the offline dataset. Lower is better for both metrics.

| Task | FISOR | | LSPC | | GSCO (ours) | |
|---|---|---|---|---|---|---|
| | OOD rate ↓ | OOD dist ↓ | OOD rate ↓ | OOD dist ↓ | OOD rate ↓ | OOD dist ↓ |
| BallCircle | 0.146 | 0.203 | 0.014 | 0.188 | 0.005 | 0.127 |
| CarPush | 0.071 | 0.303 | 0.035 | 0.246 | 0.019 | 0.195 |
| MediumDense | 0.113 | 0.345 | 0.086 | 0.351 | 0.069 | 0.304 |

**Direct OOD Measurement.** To directly assess whether geometric regularization reduces practical action-space shift, we introduce two state-conditioned OOD diagnostics computed against the feasible support of the offline dataset. We define the feasible support as transitions from zero-violation trajectories. For each evaluation state $s$ and policy action $a = \pi(s)$, we first retrieve the $k$ nearest feasible dataset states and compute

$$d_{\mathcal{D}}^{\text{feas}}(s, a) = \min_{(s_i, a_i) \in \mathcal{N}_k^{\text{feas}}(s)} \|\bar{a} - a_i\|_2,$$

where $\bar{a}$ and $a_i$ are normalized actions. We report the mean feasible-support distance and the OOD action rate

$$\Pr\left[d_{\mathcal{D}}^{\text{feas}}(s, \pi(s)) > \tau_{\text{feas}}\right],$$

where $\tau_{\text{feas}}$ is calibrated as a high quantile (0.95) of the same local feasible-support distance distribution. Lower values indicate that generated actions stay closer to the local support of the offline data.

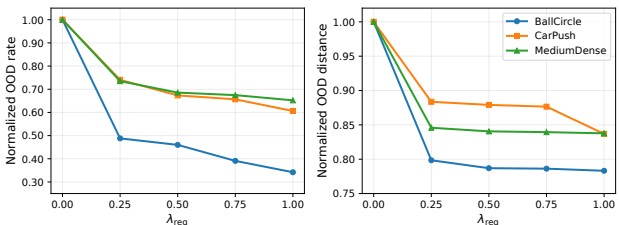

*Figure 5.* Effect of geometric regularization on direct OOD measurements. (a) Normalized OOD rate versus $\lambda_{\text{reg}}$. (b) Normalized OOD distance versus $\lambda_{\text{reg}}$. Each curve is normalized by its value at $\lambda_{\text{reg}} = 0$. Compared with the unregularized variant, adding geometric regularization markedly reduces measured action-space shift, while larger $\lambda_{\text{reg}}$ values lead to smaller additional gains with a mild overall downward trend.

Table 6 compares GSCO with two competitive generative safe offline RL baselines, FISOR and LSPC. GSCO attains the lowest OOD rate and OOD distance on all representative tasks, suggesting that the geometric refinement keeps generated actions closer to the local support of the offline data. Figure 5 further shows that increasing $\lambda_{\text{reg}}$ consistently reduces measured shift, with the largest improvement obtained when the regularizer is first introduced. Together, these results empirically validate the role of $L_{\text{reg}}$ as a lightweight mechanism for suppressing action-space drift during refinement.

## 7. Conclusion

We presented GSCO, a geometric framework for safe offline reinforcement learning that regulates policy improvement through a bijective latent representation. By learning a feasibility-aware flow prior and performing structured refinement in the base space, GSCO decouples reward improvement, safety correction, and geometric regularization. Our analysis identifies base-space divergence as a theoretical control target for downstream policy shift, while the implemented regularizer provides a lightweight surrogate for suppressing practical action-space drift. Experiments across standard safe offline RL benchmarks show that GSCO achieves low violation rates and competitive reward–safety trade-offs.

Several limitations remain. First, GSCO relies on feasibility estimation from purely offline data, which can be challenging when cost signals are sparse or safety boundaries are complex. Second, the current evaluation focuses on low-dimensional state benchmarks; extending the method to image-based or other high-dimensional observations would require learned representations whose quality may affect both feasibility estimation and geometric regularization. Third, in harder driving tasks such as Safe MetaDrive, the limited overlap between high-reward and low-cost behaviors can make the method conservative. Future work includes adaptive refinement schedules, more principled manifold-shaping objectives, and scalable representation learning for high-dimensional safe offline RL.

## Impact Statement

This paper presents work whose goal is to advance the field of machine learning, particularly in safe offline reinforcement learning. There are many potential societal consequences of this line of research, including both positive applications in safety-critical decision-making systems and potential misuse if deployed without appropriate safeguards. While our method is designed to reduce out-of-distribution policy shift and safety violations in offline learning, it should not be viewed as a substitute for system-level validation, monitoring, and domain-specific safety certification before real-world deployment. We do not believe that this work raises specific ethical concerns beyond those commonly associated with reinforcement learning research.

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

# A. Extended Discussions on Related Works

## A.1. Generative Latent-Space Offline RL Methods

A growing line of offline RL methods learns policies in a low-dimensional latent action or trajectory space induced by a generative model. These approaches typically fit a conditional generative model on offline data and optimize a latent policy whose outputs are decoded back to actions, thereby constraining policy search to a data-supported manifold and reducing OOD actions. PLAS (Zhou et al., 2021) and its CVAE-based extensions, such as LAPO (Chen et al., 2022) and ELAPSE (Han et al., 2025), enhance this framework by shaping the latent distribution to emphasize high-return behaviors and mitigate collapse. In the model-based setting, C-LAP (Alles et al., 2024) learns a latent action state-space model and constrains imagined rollouts to remain within the latent prior, providing implicit conservatism. Latent diffusion approaches (Venkatraman et al., 2023) extend this idea to trajectory-level latent spaces, enabling policy optimization over semantically structured latent trajectories. Flow-based generative policies have also been explored; CNF (Akimov et al., 2022) trains a normalizing flow over actions and reduces OOD actions by bounding the base distribution under a frozen decoder. CPED (Zhang et al., 2023a) explicitly estimates the behavior density using a flow-GAN and constrains policy updates within high-density regions. For safe offline RL, LSPC (Koirala et al., 2024) encodes latent safety constraints with a CVAE and regularizes the latent policy using a safety critic, though it still relies on ELBO training and support-based constraints. Prior-guided diffusion planning (PG) (Ki et al., 2026) is also closely related in motivation, as it mitigates offline RL distributional shift by guiding diffusion-based planning with a learned prior in latent space. However, PG is designed for general offline RL and primarily regularizes diffusion sampling through the learned prior. In contrast, GSCO targets safe offline RL, where policy improvement must jointly balance reward, feasibility, and support preservation. Moreover, GSCO uses an invertible flow with exact likelihood and a base-space geometric regularizer, yielding a direct mechanism for monitoring and suppressing action-space shift during refinement.

Compared with these generative latent(-space) policy methods, GSCO differs along four complementary dimensions, summarized here and in Table 1 in the main text.

1. **Task scope and safety objective.** Prior flow-based methods such as CNF (Akimov et al., 2022) do not target safe offline RL. GSCO lies in the same flow-based family but is instantiated for hard-constrained safe offline RL with near-zero violation, rather than for unconstrained or budgeted objectives.

2. **Generative backbone and likelihood.** CVAE-, flow-, and diffusion-based policies all exploit latent manifolds, but flows are invertible and admit exact likelihoods. With a Gaussian base, GSCO can monitor a base-space KL divergence and propagate it into bounds on action/policy deviation (TV/$W_2$) and OOD mass, providing an analyzable and tunable surrogate notion of conservatism not available to ELBO-trained CVAEs (Zhou et al., 2021; Chen et al., 2022; Han et al., 2025) or multi-step latent diffusion models (Venkatraman et al., 2023). This also differs from prior-guided diffusion planning (Ki et al., 2026), which guides an implicit diffusion sampler with a learned prior rather than using an invertible likelihood model.

3. **OOD-control mechanism.** CNF reduces OOD by making the flow's base uniform-bounded and freezing the decoder (Akimov et al., 2022), but does not explicitly control policy deviation. GSCO instead (a) retains a Gaussian base whose divergence provides an analyzable control target for policy shift and (b) performs feasibility-guided density shaping on the base (using the flow's inverse). Together, this makes conservatism measurable and systematically regulatable, while keeping policy search within empirically safe, high-density regions.

4. **Training and inference protocol.** Safe offline RL couples reward, safety, and OOD control. Instead of relying on a single entangled loss, GSCO employs ordered small-step refiners in the base space with a frozen decoder—Safety $\rightarrow$ Reward $\rightarrow$ Shared—so updates remain in-support and locally constrained. This protocol tightly links safety, reward, and OOD suppression, exposes a clear trade-off handle, and avoids the instability of lumping all terms into one gradient.

## A.2. Additional Discussion on Hard and Soft Constraint

**Hard vs. soft formulations.** In safe RL, a *hard* (state-wise) safety constraint requires that the policy never leaves the safe set. Let $h : \mathcal{S} \rightarrow \mathbb{R}$ encodes a state constraint and $c(s) = \max\{h(s), 0\}$ is the induced cost. A hard constraint enforces

$$h(s_t) \leq 0, \quad a_t \sim \pi(\cdot \mid s_t), \ \forall t \in \mathbb{N}, \tag{17}$$

which can equivalently be written as a zero-violation cost condition

$$c(s_t) = 0, \quad a_t \sim \pi(\cdot \mid s_t), \ \forall t \in \mathbb{N}. \tag{18}$$

By contrast, *soft* or *budgeted* constraints are typically expressed at the level of expected cumulative cost. Given a cost limit $l > 0$, the constraint is

$$\mathbb{E}_{\tau \sim \pi}\Big[ \sum_{t=0}^{\infty} c(s_t) \Big] \le l \quad \text{or} \quad \mathbb{E}_{\tau \sim \pi}\Big[ \sum_{t=0}^{\infty} \gamma^t c(s_t) \Big] \le l, \tag{19}$$

and the policy is allowed to incur nonzero instantaneous violations as long as the long-term budget is respected. Recent work further extends this perspective to *real-time* budgeted safety, where the agent must adapt to dynamically specified cost budgets in the offline setting (Lin et al., 2023), as well as to risk- and distributionally-robust variants (Chow et al., 2019; Kushwaha et al., 2025).

**Design philosophies and use cases.**   Hard and soft formulations reflect different safety philosophies rather than a strict ordering of capability. Hard/near-zero-violation methods (Fisac et al., 2018; Yu et al., 2022; Zheng et al., 2024; Zhao et al., 2023) target scenarios where every violation corresponds to an unacceptable safety breach (e.g., collisions, irreversible damage, or regulatory violations); here, the emphasis is on characterizing and staying inside the feasible region. Budgeted or soft methods (Le et al., 2019; Lee et al., 2022; Liu et al., 2023b), in contrast, model cost as an allocatable resource: small, occasional violations are acceptable if they enable substantially better task performance, which is appropriate for risk-sensitive but non-safety-critical domains or applications with tunable risk budgets.

Our framework intentionally follows the hard / near-zero-violation viewpoint: we are interested in safe offline RL settings where violations correspond to genuine safety failures, and thus focus on maximizing return while keeping state-wise safety rates close to $100\%$. We view budgeted-safety approaches as complementary rather than competing; in principle, similar generative latent-space and flow-based techniques could be adapted to budgeted formulations by conditioning critics and refiners on a dynamic cost budget, which we leave as an interesting direction for future work.

# B. Theoretical Analysis

In this section, we provide the missing proofs for the theoretical results to support or validate the proposed method.

## B.1. Derivation of the Flow density

Normalizing flows model complex distributions by transporting samples from a simple base density through an invertible transformation. In the conditional setting, let $u \sim p_0(u)$ denote a latent variable drawn from a base distribution, typically $\mathcal{N}(0, I)$, and define $z = f_\phi(u; \text{cond})$, where $f_\phi(\cdot; \text{cond})$ is a bijective mapping parameterized by $\phi$ and conditioned on an external variable $\text{cond}$ (e.g., a state or context).

Because the map is invertible for fixed $\text{cond}$, the inverse $u = f_\phi^{-1}(z; \text{cond})$ is well defined. To obtain the conditional density $p_\phi(z \mid \text{cond})$, we apply the change-of-variables formula for differentiable bijections:

$$p(z) = p_0\big(f^{-1}(z)\big) \cdot \left| \det \frac{\partial f^{-1}(z)}{\partial z} \right| = p_0(u) \cdot \left| \det \frac{\partial f(u)}{\partial u} \right|^{-1}, \tag{20}$$

where the second equality follows from the inverse function theorem. In our conditional setting we thus have

$$p_\phi(z \mid \text{cond}) = p_0(u) \cdot \left| \det \frac{\partial u}{\partial z} \right|, \quad u = f_\phi^{-1}(z; \text{cond}). \tag{21}$$

Using $\frac{\partial u}{\partial z} = \big(\frac{\partial z}{\partial u}\big)^{-1}$, we can express the inverse Jacobian in terms of the forward transformation:

$$\left| \det \frac{\partial u}{\partial z} \right| = \left| \det \frac{\partial f_\phi(u; \text{cond})}{\partial u} \right|^{-1}. \tag{22}$$

Substituting this identity back into the density expression gives

$$p_\phi(z \mid \text{cond}) = p_0(u) \cdot \left| \det \frac{\partial f_\phi(u; \text{cond})}{\partial u} \right|^{-1}, \quad u = f_\phi^{-1}(z; \text{cond}). \tag{23}$$

Taking logarithms yields the exact log-likelihood of the conditional flow:

$$\log p_\phi(z \mid \text{cond}) = \log p_0(u) + \log \left| \det \frac{\partial u}{\partial z} \right|, \quad u = f_\phi^{-1}(z; \text{cond}), \tag{2}$$

which corresponds to Eq. 2 in the main text.

In practice, the Jacobian determinant is computed analytically using affine coupling layers, whose triangular structure reduces the log-determinant to a sum of layerwise log-scale outputs. This makes the likelihood term efficient to compute while preserving the exactness afforded by the invertibility of the flow.

When the transformation is a composition of $L$ conditional bijections,

$$u_0 \sim p_0, \quad u_\ell = f_\ell(u_{\ell-1}; \text{cond}), \; \ell = 1, \ldots, L, \quad z = u_L, \tag{24}$$

The change-of-variables formula yields

$$\log p_\phi(z \mid \text{cond}) = \log p_0(u_0) + \sum_{\ell=1}^{L} \log \left| \det \frac{\partial u_{\ell-1}}{\partial u_\ell} \right|, \tag{25}$$

where each term uses the inverse Jacobian of layer $f_\ell$. Equivalently, this can be written as the negative sum of forward log-determinants,

$$\log p_\phi(z \mid \text{cond}) = \log p_0(u_0) - \sum_{\ell=1}^{L} \log \left| \det \frac{\partial f_\ell(u_{\ell-1}; \text{cond})}{\partial u_{\ell-1}} \right|, \tag{26}$$

which is the form implemented in practice when accumulating the density term across multiple flow layers.

## B.2. Feasibility-based Value Function

The state-wise zero-violation requirement in Eq. 1 calls for a trajectory-level safety signal rather than a purely per-step expected cost. Hamilton–Jacobi (HJ) reachability (Bansal et al., 2017) from safe control provides exactly such a representation through signed safety functions and value-based certificates, and has been shown to be effective for enforcing hard constraints in recent safe RL studies (Fisac et al., 2018; Yu et al., 2022). Following this line, we cast the hard constraint into a pair of feasibility value functions based on Definition B.1 that we can learn from offline data and then use as a unified signal for policy generation and refinement.

**Definition B.1** (Optimal feasible value functions). Let $h : \mathcal{S} \to \mathbb{R}$ be a signed safety function with $h(s) \leq 0$ denoting safety, and let $T(\cdot|s, a)$ denote the transition kernel. For a policy $\pi$, trajectories are generated by $a_t \sim \pi(\cdot|s_t)$ and $s_{t+1} \sim T(\cdot|s_t, a_t)$. The optimal state-wise and action-wise feasibility values are defined by

$$V_h^\star(s) := \min_\pi \mathbb{E}_{\pi, T} \left[ \max_{t \in \mathbb{N}} h(s_t) \mid s_0 = s \right], \tag{27}$$

$$Q_h^\star(s, a) := \min_\pi \mathbb{E}_{\pi, T} \left[ \max_{t \in \mathbb{N}} h(s_t) \mid s_0 = s, \, a_0 = a, \, a_{t \geq 1} \sim \pi(\cdot|s_t) \right]. \tag{28}$$

Here the expectation is over policy randomness and stochastic transitions. Thus, $V_h^\star(s) \leq 0$ indicates the existence of a policy whose expected worst-case violation from $s$ is non-positive, and $Q_h^\star(s, a) \leq 0$ gives the corresponding action-wise feasibility condition. In offline settings, these quantities can be approximated by a discounted feasible Bellman operator.

**Definition B.2** (Feasible Bellman operator). For $\gamma \in (0, 1)$ and any $Q : \mathcal{S} \times \mathcal{A} \to \mathbb{R}$, define $V_Q(s) := \min_a Q(s, a)$. The feasible Bellman operator is

$$(\mathcal{P}^\star Q)(s, a) := (1 - \gamma) h(s) + \gamma \, \mathbb{E}_{s' \sim T(\cdot|s, a)} \left[ \max\{h(s), V_Q(s')\} \right]. \tag{29}$$

This operator is a $\gamma$-contraction under the sup norm and admits a unique fixed point $Q_{h,\gamma}^\star$, with $V_{h,\gamma}^\star(s) = \min_a Q_{h,\gamma}^\star(s, a)$. As $\gamma \uparrow 1$, the discounted fixed point approaches the HJ-style feasibility values in Definition B.1 under standard regularity assumptions. The proof is deferred to Appendix B.2.1.

We parameterize $(Q_h, V_h)$ with neural networks. To avoid extrapolation errors that arise from querying actions outside the data support (Fujimoto et al., 2019), we approximate $Q_h(s, \cdot)$ by reversed expectile regression and train $Q_h$ with a one-step target that uses $V_h$ in place of $\min_{a'} Q_h(s', a')$:

$$\mathcal{L}_{V_h} = \mathbb{E}_{(s,a)\sim\mathcal{D}}\left[\rho_{\tau_h}^{\text{rev}}\left(Q_h(s,a) - V_h(s)\right)\right], \tag{30}$$

$$\mathcal{L}_{Q_h} = \mathbb{E}_{(s,a,s')\sim\mathcal{D}}\left[\left((1-\gamma)h(s) + \gamma\max\{h(s), V_h^{\text{tgt}}(s')\} - Q_h(s,a)\right)^2\right]. \tag{31}$$

where $\rho_\tau^{\text{rev}}(u) = |\tau - \mathbf{1}\{u > 0\}|\, u^2$ and $V_h^{\text{tgt}}$ is a slowly updated target network. The reversed expectile with $\tau_h \in (0.5, 1)$ down-weights overly optimistic $Q_h$ values and sharpens the zero level set $V_h \approx 0$, while the target network stabilizes bootstrapping.

### B.2.1. PROOF OF DEFINITION B.2.

For a fixed $\gamma \in (0, 1)$ and we define $V_i(s) := \min_a Q_i(s, a)$ for $i \in \{1, 2\}$. Then for any $(s, a)$,

$$\left|(\mathcal{P}^\star Q_1)(s,a) - (\mathcal{P}^\star Q_2)(s,a)\right| = \gamma\left|\mathbb{E}_{s'}\left[\max\{h(s), V_1(s')\} - \max\{h(s), V_2(s')\}\right]\right| \tag{32}$$
$$\leq \gamma\,\mathbb{E}_{s'}\left|V_1(s') - V_2(s')\right|.$$

Since $V_i(s') = \min_{a'} Q_i(s', a')$ and the pointwise min is 1-Lipschitz, $\left|V_1(s') - V_2(s')\right| \leq \sup_{a'} |Q_1(s', a') - Q_2(s', a')| \leq \|Q_1 - Q_2\|_\infty$. Taking the supremum over $(s, a)$ yields

$$\|\mathcal{P}^\star Q_1 - \mathcal{P}^\star Q_2\|_\infty \leq \gamma\|Q_1 - Q_2\|_\infty, \tag{33}$$

so $\mathcal{P}^\star$ is a $\gamma$-contraction under the sup norm. By Banach's fixed-point theorem, there exists a unique fixed point $Q_{h,\gamma}^\star$ and we set $V_{h,\gamma}^\star(s) := \min_a Q_{h,\gamma}^\star(s, a)$.

To connect to the undiscounted HJ-style values, assume $h$ is bounded. Let $\gamma_n \uparrow 1$ and consider the fixed points $Q_{h,\gamma_n}^\star$. Because $\{Q_{h,\gamma_n}^\star\}_n$ is uniformly bounded and $\mathcal{P}^\star$ is continuous in $\gamma$, any limit point $Q^\dagger$ satisfies, for all $(s, a)$,

$$Q^\dagger(s,a) = \lim_{n\to\infty}\left[(1-\gamma_n)h(s) + \gamma_n\,\mathbb{E}_{s'}\left[\max\{h(s), \min_{a'} Q_{h,\gamma_n}^\star(s', a')\}\right]\right] \tag{34}$$
$$= \mathbb{E}_{s'}\left[\max\{h(s), \min_{a'} Q^\dagger(s', a')\}\right].$$

This is the dynamic programming equation for the HJ-style (statewise zero-violation) feasibility values; hence $Q^\dagger = Q_h^\star$ and $V^\dagger = \min_a Q^\dagger(\cdot, a) = V_h^\star$. Therefore $Q_{h,\gamma}^\star \to Q_h^\star$ and $V_{h,\gamma}^\star \to V_h^\star$ as $\gamma \uparrow 1$. $\qquad\square$

### B.3. Proof of Lemma 4.1

Recall the feasibility-weighted variational objective:

$$\mathcal{L}_{\text{ELBO}} = \mathbb{E}_{(s,a)\sim p_\mathcal{D}}\left[w(s,a)\mathbb{E}_{z\sim q_\psi(z|s,a)}[-\log \pi_\theta(a|s,z)]\right] \tag{35}$$
$$+ \beta\,\mathbb{E}_{(s,a)\sim p_\mathcal{D}}[w(s,a)D_{\text{KL}}(q_\psi(\cdot|s,a)\|p_\phi(\cdot|s))].$$

Define the normalized feasibility-weighted empirical distribution

$$\tilde{p}_\mathcal{D}(s,a) = \frac{w(s,a)p_\mathcal{D}(s,a)}{Z_w}, \qquad Z_w = \mathbb{E}_{p_\mathcal{D}}[w(s,a)].$$

Let $\tilde{p}_\mathcal{D}(s)$ be its state marginal and $\tilde{p}_\mathcal{D}(a|s) = \tilde{p}_\mathcal{D}(s,a)/\tilde{p}_\mathcal{D}(s)$. For the case $\beta = 1$, dividing by the positive constant $Z_w$ gives

$$Z_w^{-1}\mathcal{L}_{\text{ELBO}} = \mathbb{E}_{(s,a)\sim\tilde{p}_\mathcal{D}}\mathbb{E}_{z\sim q_\psi}[-\log \pi_\theta(a|s,z)]$$
$$+ \mathbb{E}_{(s,a)\sim\tilde{p}_\mathcal{D}}[D_{\text{KL}}(q_\psi(\cdot|s,a)\|p_\phi(\cdot|s))] \tag{36}$$
$$= \mathbb{E}_{(s,a)\sim\tilde{p}_\mathcal{D}}\mathbb{E}_{z\sim q_\psi}\left[\log\frac{q_\psi(z|s,a)}{p_\phi(z|s)\pi_\theta(a|s,z)}\right].$$

Now define two joint distributions over $(s, a, z)$:

$$P_\psi(s, a, z) = \tilde{p}_\mathcal{D}(s, a) q_\psi(z|s, a),$$

$$Q_{\phi,\theta}(s, a, z) = \tilde{p}_\mathcal{D}(s) p_\phi(z|s) \pi_\theta(a|s, z).$$

Both are normalized joint distributions, assuming $\pi_\theta(\cdot|s, z)$ denotes the normalized reconstruction likelihood, or the corresponding conditional measure induced by the decoder. Then

$$
\begin{aligned}
D_{\mathrm{KL}}(P_\psi \| Q_{\phi,\theta}) &= \mathbb{E}_{P_\psi}\left[\log \frac{\tilde{p}_\mathcal{D}(s, a) q_\psi(z|s, a)}{\tilde{p}_\mathcal{D}(s) p_\phi(z|s) \pi_\theta(a|s, z)}\right] \\
&= \mathbb{E}_{P_\psi}\left[\log \frac{q_\psi(z|s, a)}{p_\phi(z|s) \pi_\theta(a|s, z)}\right] + \mathbb{E}_{\tilde{p}_\mathcal{D}}\left[\log \tilde{p}_\mathcal{D}(a|s)\right].
\end{aligned}
\tag{37}
$$

Therefore,

$$Z_w^{-1} \mathcal{L}_{\mathrm{ELBO}} = D_{\mathrm{KL}}(P_\psi \| Q_{\phi,\theta}) - \mathbb{E}_{\tilde{p}_\mathcal{D}}\left[\log \tilde{p}_\mathcal{D}(a|s)\right]. \tag{38}$$

The second term depends only on the fixed weighted empirical distribution and is independent of $(\phi, \psi, \theta)$. Thus, for fixed weights and $\beta = 1$, minimizing $\mathcal{L}_{\mathrm{ELBO}}$ is equivalent, up to a positive scale factor and a model-independent additive constant, to minimizing the valid matched-joint KL divergence $D_{\mathrm{KL}}(P_\psi \| Q_{\phi,\theta})$.

For $\beta \neq 1$, the objective can be decomposed as

$$
\begin{aligned}
Z_w^{-1} \mathcal{L}_{\mathrm{ELBO}} = {}& D_{\mathrm{KL}}(P_\psi \| Q_{\phi,\theta}) - \mathbb{E}_{\tilde{p}_\mathcal{D}}\left[\log \tilde{p}_\mathcal{D}(a|s)\right] \\
& + (\beta - 1)\mathbb{E}_{(s,a)\sim\tilde{p}_\mathcal{D}}\left[D_{\mathrm{KL}}(q_\psi(\cdot|s, a) \| p_\phi(\cdot|s))\right].
\end{aligned}
\tag{39}
$$

Hence, for general $\beta$, the objective should be interpreted as a feasibility-weighted variational loss with a scaled posterior-prior regularizer, rather than as an exact matched-joint KL projection. This clarification affects only the interpretation of the objective and does not change the algorithm or empirical results. $\qquad\square$

### B.4. Proof of Lemma 4.2

We prove the lemma in two steps: first by decomposing the KL divergence using the chain rule, and second by bounding the geometric divergence using the properties of the normalizing flow and the frozen decoder. We assume that, for each state $s$, the pushforward prior policy $\pi_0(\cdot|s)$ is absolutely continuous with respect to the behavior policy $\pi_\beta(\cdot|s)$ on the effective support of interest, and that there exists a finite constant $R_\theta(s) < \infty$ such that

$$\frac{\pi_0(a|s)}{\pi_\beta(a|s)} \leq R_\theta(s), \quad \forall a \in \mathrm{supp}(\pi_0(\cdot|s)). \tag{40}$$

Under this assumption, the mismatch term in Lemma 4.2 satisfies $C(s) = \log R_\theta(s)$.

**Step 1: Decomposition.** Let $\pi = \Pi_\theta(\cdot|s)$, $\pi_0 = \pi_0(\cdot|s)$, and $\pi_\beta = \pi_\beta(\cdot|s)$ denote the refined policy, the base policy (flow prior), and the behavior policy, respectively. Using the chain rule for KL divergence (assuming absolute continuity and bounded support), we have:

$$D_{\mathrm{KL}}(\pi \| \pi_\beta) = D_{\mathrm{KL}}(\pi \| \pi_0) + \mathbb{E}_{a\sim\pi}\left[\log \frac{\pi_0(a)}{\pi_\beta(a)}\right]. \tag{41}$$

Let $R_\theta(s) := \sup_a \frac{\pi_0(a|s)}{\pi_\beta(a|s)}$ be the supremum density ratio on the data support. We can bound the second term:

$$\mathbb{E}_{a\sim\pi}\left[\log \frac{\pi_0(a)}{\pi_\beta(a)}\right] \leq \log R_\theta(s) =: C(s). \tag{42}$$

Thus, $D_{\mathrm{KL}}(\pi \| \pi_\beta) \leq D_{\mathrm{KL}}(\pi \| \pi_0) + C(s)$.

**Step 2: Geometric Bound via DPI.** Now we bound the term $D_{\mathrm{KL}}(\pi \,\|\, \pi_0)$. Recall our generative process: $u \to z = f_\phi(u) \to a = T_s(z)$. * **Bijectivity:** Since $f_\phi$ is a diffeomorphism (bijective flow), the KL divergence is invariant under this transformation: $D_{\mathrm{KL}}(q_z \,\|\, p_\phi) = D_{\mathrm{KL}}(q_u \,\|\, \mathcal{N})$. * **Data Processing Inequality (DPI):** The decoder $T_s$ is a deterministic (or stochastic) mapping. By the Data Processing Inequality for $f$-divergences (Csiszár & Shields, 2004), applying a mapping cannot increase the divergence:

$$D_{\mathrm{KL}}(\pi \,\|\, \pi_0) \;=\; D_{\mathrm{KL}}(T_{s\#}q_z \,\|\, T_{s\#}p_\phi) \;\leq\; D_{\mathrm{KL}}(q_z \,\|\, p_\phi). \tag{43}$$

Combining Step 1 and Step 2, and substituting $D_{\mathrm{KL}}(q_z \,\|\, p_\phi) = D_{\mathrm{KL}}(q_u \,\|\, \mathcal{N})$, we obtain:

$$D_{\mathrm{KL}}(\pi \,\|\, \pi_\beta) \;\leq\; D_{\mathrm{KL}}(q_u \,\|\, \mathcal{N}) + C(s). \tag{44}$$

$\square$

### B.5. Proof of Corollary 4.3

We prove the bounds for the Wasserstein distance and the Total Variation (TV) divergence separately. Let $u \sim q_u$ be the refined base distribution and $u_0 \sim \mathcal{N}(0, I)$ be the prior. Let $z = f_\phi(u; s)$ and $a = g_\theta(z; s)$ denote the flow and decoder mappings, respectively.

**1. Wasserstein Bound.** Let $\pi = g_{\theta\#}(f_{\phi\#}q_u)$ and $\pi_0 = g_{\theta\#}(f_{\phi\#}\mathcal{N})$. First, consider the latent space distributions $q_z = f_{\phi\#}q_u$ and $p_\phi = f_{\phi\#}\mathcal{N}$. Since the flow $f_\phi$ is a diffeomorphism, the KL divergence is invariant under the change of variables:

$$D_{\mathrm{KL}}(q_z\|p_\phi) \;=\; D_{\mathrm{KL}}(q_u\|\mathcal{N}). \tag{45}$$

Next, if the decoder $g_\theta(\cdot; s)$ is $L_g$-Lipschitz continuous on the latent manifold, then the pushforward operation implies that the 2-Wasserstein distance satisfies:

$$W_2(\pi, \pi_0) \;\leq\; L_g\, W_2(q_z, p_\phi). \tag{46}$$

By Talagrand's $T_2$ inequality (which holds for the Gaussian reference measure or distributions satisfying a log-Sobolev inequality) (Otto & Villani, 2000), we can bound the latent Wasserstein distance by the KL divergence:

$$W_2(q_z, p_\phi) \;\leq\; \sqrt{2\, D_{\mathrm{KL}}(q_z\|p_\phi)}. \tag{47}$$

Combining these results yields the final bound:

$$W_2(\pi, \pi_0) \;\leq\; L_g\sqrt{2\, D_{\mathrm{KL}}(q_u\|\mathcal{N})}. \tag{48}$$

**2. Total Variation and OOD Probability.** By the triangle inequality, the deviation from the behavior policy is bounded by:

$$\mathrm{TV}(\pi, \pi_\beta) \;\leq\; \mathrm{TV}(\pi, \pi_0) + \mathrm{TV}(\pi_0, \pi_\beta). \tag{49}$$

To bound the first term, we use Pinsker's inequality:

$$\mathrm{TV}(\pi, \pi_0) \;\leq\; \sqrt{\tfrac{1}{2}D_{\mathrm{KL}}(\pi\|\pi_0)}. \tag{50}$$

By the Data Processing Inequality (DPI) for $f$-divergences, applying the deterministic decoder mapping $g_\theta$ cannot increase the KL divergence. Combined with the flow invariance (Eq. 45), we have:

$$D_{\mathrm{KL}}(\pi\|\pi_0) \;\leq\; D_{\mathrm{KL}}(q_z\|p_\phi) \;=\; D_{\mathrm{KL}}(q_u\|\mathcal{N}). \tag{51}$$

Thus, $\mathrm{TV}(\pi, \pi_0) \leq \sqrt{\tfrac{1}{2}D_{\mathrm{KL}}(q_u\|\mathcal{N})}$. Finally, for any measurable OOD region $\mathcal{O}$, by the definition of total variation distance, $\pi(\mathcal{O}) - \pi_\beta(\mathcal{O}) \leq \mathrm{TV}(\pi, \pi_\beta)$. Substituting the bounds completes the proof:

$$\pi(\mathcal{O}) \;\leq\; \pi_\beta(\mathcal{O}) + \sqrt{\tfrac{1}{2}D_{\mathrm{KL}}(q_u\|\mathcal{N})} + \mathrm{TV}(\pi_0, \pi_\beta). \tag{52}$$

**Remark on Lipschitz Continuity.** The Wasserstein bound relies on a (local) Lipschitz constant $L_g$ of the composite decoder–flow mapping on the learned manifold. Such a constant always exists on any compact subset by continuity of neural networks. In principle, $L_g$ can be upper-bounded or regularized via standard techniques in generative modeling, such as spectral normalization (Miyato et al., 2018) or gradient penalties (Gulrajani et al., 2017), although we do not attempt to numerically certify this quantity in this work. Accordingly, the bound is used to characterize the monotonic dependence of downstream policy deviation on the base-space divergence $D_{\mathrm{KL}}(q_u\|\mathcal{N})$, rather than to compute certified numerical deviation radii.

**Independence of Update Order.** We note that the derived bounds depend solely on the divergence of the *final* refined base distribution $q_u$, regardless of the specific optimization trajectory or the order of expert updates used to reach it. Consequently, the theoretical guarantees hold for any valid refinement scheme (sequential, parallel, or averaged) provided the final latent energy is bounded.

### B.6. Policy Gap and Comparison with Prior Performance Bounds

We derive explicit performance bounds that relate the reward and cost gaps between the refined policy and the flow prior (or behavior policy) to the base-space KL regularizer used in GSCO. These results formally justify minimizing the base-space energy.

**Preliminaries.** Let $\pi$ denote the final refined policy, $\pi_0$ the flow prior policy (base policy), and $\pi_\beta$ the behavior policy. We assume rewards and costs are bounded as $|r(s,a)| \leq R_{\max}$ and $|c(s,a)| \leq C_{\max}$. We write $J_r(\pi) := \mathbb{E}[\sum_{t\geq 0} \gamma^t r_t]$ and $J_h(\pi) := \mathbb{E}[\sum_{t\geq 0} \gamma^t c_t]$ for the expected reward and cost return under $\pi$. Let $d^\pi_{\rho_0}$ be the discounted state-visitation distribution. For a reference policy $\pi'$, we define the advantage functions $A^{\pi'}_r(s,a) = Q^{\pi'}_r(s,a) - V^{\pi'}_r(s)$ and $A^{\pi'}_h(s,a) = Q^{\pi'}_h(s,a) - V^{\pi'}_h(s)$.

**Lemma B.3** (Performance difference via TV). *For any two policies $\pi$ and $\pi'$, the performance difference is bounded by the total variation (TV) divergence:*

$$\left| J_r(\pi) - J_r(\pi') \right| \;\leq\; \frac{2R_{\max}}{(1-\gamma)^2} \sup_s \mathrm{TV}\big(\pi(\cdot|s), \pi'(\cdot|s)\big). \tag{53}$$

*An analogous bound holds for the safety cost $J_h$ with $R_{\max}$ replaced by $C_{\max}$.*

*Proof.* The equalities are the standard performance-difference lemma obtained by unrolling the Bellman equations and telescoping the resulting series.

For the inequality, bounded rewards imply $|V^{\pi'}_r(s)| \leq R_{\max}/(1-\gamma)$ and $|Q^{\pi'}_r(s,a)| \leq R_{\max}/(1-\gamma)$ for all $(s,a)$, hence $\left| A^{\pi'}_r(s,a) \right| \leq 2R_{\max}/(1-\gamma)$. Moreover, for every $s$ we have $\mathbb{E}_{a\sim\pi'(\cdot|s)}[A^{\pi'}_r(s,a)] = 0$, so

$$\left| \mathbb{E}_{a\sim\pi(\cdot|s)} A^{\pi'}_r(s,a) \right| = \left| \mathbb{E}_{a\sim\pi(\cdot|s)} A^{\pi'}_r(s,a) - \mathbb{E}_{a\sim\pi'(\cdot|s)} A^{\pi'}_r(s,a) \right| \leq 2\frac{R_{\max}}{1-\gamma} \mathrm{TV}\big(\pi(\cdot|s), \pi'(\cdot|s)\big), \tag{54}$$

where we used the standard inequality $|\mathbb{E}_p f - \mathbb{E}_q f| \leq 2\|f\|_\infty \mathrm{TV}(p,q)$. Plugging this bound into the performance-difference lemma and taking the supremum over $s$ yields

$$\left| J_r(\pi) - J_r(\pi') \right| \leq \frac{1}{1-\gamma} \mathbb{E}_{s\sim d^\pi_{\rho_0}} qu\Big[ 2\frac{R_{\max}}{1-\gamma} \mathrm{TV}\big(\pi(\cdot|s), \pi'(\cdot|s)\big) \Big] \leq \frac{2R_{\max}}{(1-\gamma)^2} \sup_s \mathrm{TV}\big(\pi(\cdot|s), \pi'(\cdot|s)\big). \tag{55}$$

The bound for $J_h$ follows by replacing $R_{\max}$ with $C_{\max}$. $\qquad\square$

**Proposition B.4** (Policy gap under base-space KL control). *Assume that the refined base latent distribution $q_u(\cdot|s)$ satisfies a uniform KL constraint against the Gaussian prior:*

$$D_{\mathrm{KL}}\big(q_u(\cdot|s) \,\|\, \mathcal{N}(\cdot)\big) \;\leq\; \varepsilon_{\mathrm{base}} \qquad \textit{for all } s, \tag{56}$$

*where $\varepsilon_{\text{base}}$ is the base-space KL radius. Let $\Delta_\beta := \sup_s \text{TV}\big(\pi_0(\cdot|s), \pi_\beta(\cdot|s)\big)$ denote the fixed modeling mismatch between the flow prior and the behavior policy. Then the refined policy satisfies the following bounds:*

$$\big|J_r(\pi) - J_r(\pi_0)\big| \leq \frac{2R_{\max}}{(1-\gamma)^2} \sqrt{\tfrac{1}{2}\varepsilon_{\text{base}}}, \tag{P1}$$

$$\big|J_h(\pi) - J_h(\pi_0)\big| \leq \frac{2C_{\max}}{(1-\gamma)^2} \sqrt{\tfrac{1}{2}\varepsilon_{\text{base}}}, \tag{P2}$$

$$\big|J_r(\pi) - J_r(\pi_\beta)\big| \leq \frac{2R_{\max}}{(1-\gamma)^2} \big(\sqrt{\tfrac{1}{2}\varepsilon_{\text{base}}} + \Delta_\beta\big), \tag{P3}$$

$$\big|J_h(\pi) - J_h(\pi_\beta)\big| \leq \frac{2C_{\max}}{(1-\gamma)^2} \big(\sqrt{\tfrac{1}{2}\varepsilon_{\text{base}}} + \Delta_\beta\big). \tag{P4}$$

*Proof.* **1. Bounding the gap to the Flow Prior ($\pi_0$).** By Lemma B.3, it suffices to control $\sup_s \text{TV}(\pi, \pi_0)$. Recall that $\pi = T_{s\#}(f_{\phi\#}q_u)$ and $\pi_0 = T_{s\#}(f_{\phi\#}\mathcal{N})$, where $T_s$ is the frozen decoder. First, by Pinsker's inequality:

$$\text{TV}(\pi, \pi_0) \leq \sqrt{\frac{1}{2}D_{\text{KL}}(\pi\|\pi_0)}. \tag{57}$$

Next, we relate the policy KL to the base KL. Since the flow $f_\phi$ is bijective, $D_{\text{KL}}(f_{\phi\#}q_u\|f_{\phi\#}\mathcal{N}) = D_{\text{KL}}(q_u\|\mathcal{N})$. By the Data Processing Inequality (DPI), the deterministic decoder $T_s$ cannot increase the KL divergence:

$$D_{\text{KL}}(\pi\|\pi_0) = D_{\text{KL}}(T_{s\#}f_{\phi\#}q_u\|T_{s\#}f_{\phi\#}\mathcal{N}) \leq D_{\text{KL}}(q_u\|\mathcal{N}) \leq \varepsilon_{\text{base}}. \tag{58}$$

Combining these yields $\text{TV}(\pi, \pi_0) \leq \sqrt{\frac{1}{2}\varepsilon_{\text{base}}}$. Substituting this into Lemma B.3 proves (P1) and (P2).

**2. Bounding the gap to the Behavior Policy ($\pi_\beta$).** By the triangle inequality for TV distance:

$$\text{TV}(\pi, \pi_\beta) \leq \text{TV}(\pi, \pi_0) + \text{TV}(\pi_0, \pi_\beta). \tag{59}$$

From the step above, $\text{TV}(\pi, \pi_0) \leq \sqrt{\frac{1}{2}\varepsilon_{\text{base}}}$. By definition, $\text{TV}(\pi_0, \pi_\beta) \leq \Delta_\beta$. Thus, $\sup_s \text{TV}(\pi, \pi_\beta) \leq \sqrt{\frac{1}{2}\varepsilon_{\text{base}}} + \Delta_\beta$. Plugging this into Lemma B.3 proves (P3) and (P4). $\square$

**Discussion: Tunable Control vs. Uncontrolled Error.** The above results highlight the theoretical advantage of our framework. Eqs. (P1)–(P4) make the role of $\varepsilon_{\text{base}}$ transparent: by constraining the refined base distribution $q_u$ to stay within a KL ball around the Gaussian prior (via $\mathcal{L}_{\text{reg}}$), we directly bound the performance and safety shift. This provides a tunable mechanism for balancing conservatism and optimality.

In contrast, related works like LSPC (Koirala et al., 2024) derive bounds based on fixed approximation errors. For instance, LSPC shows:

$$V_r^{\pi^\star}(\rho_0) - V_r^\pi(\rho_0) \leq \frac{2R_{\max}}{(1-\gamma)^2} \left(\sqrt{\frac{\varepsilon_1'}{2}} + \sqrt{\frac{\varepsilon_2'}{2}}\right), \tag{60}$$

where $\varepsilon_1', \varepsilon_2'$ are inherent errors of the CVAE and value estimators. While similar in form, our $\varepsilon_{\text{base}}$ is a **control variable** (a regularization parameter) rather than a fixed system error. This distinction is crucial: GSCO allows practitioners to explicitly trade off conservatism against optimality by tuning the tightness of the base-space constraint, a capability grounded in the bijective geometry of the flow.

**Remark (On the KL surrogate in base-space regularization).** In practice, we do not explicitly estimate or optimize the true base-space divergence $D_{KL}(q_u\|\mathcal{N})$, which would require density estimation over the refined distribution $q_u$ and can be computationally costly and sensitive in high dimensions. Instead, we employ a lightweight density-shift surrogate induced by the invertible shared projection, consisting of a quadratic base energy, the associated change-of-variables log-determinant term, and a proximal stabilizer. This surrogate is not equivalent to the KL divergence in general. However, under mild regularity conditions (e.g., when the refined samples remain concentrated and do not exhibit extreme outliers), this surrogate provides a coarse notion of dispersion relative to the Gaussian prior and empirically stabilizes refinement. The KL-based

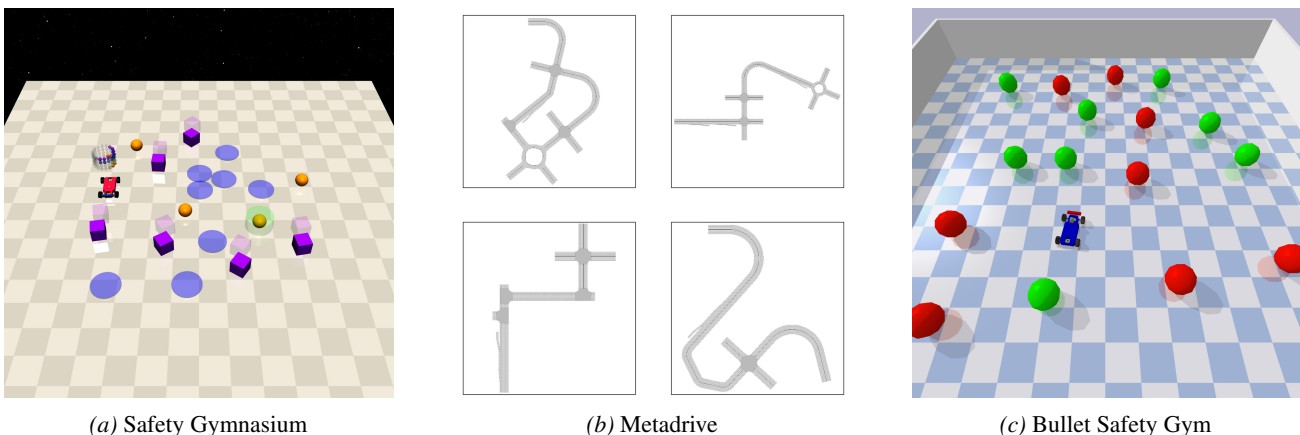

*(a)* Safety Gymnasium  *(b)* Metadrive  *(c)* Bullet Safety Gym

*Figure 6.* Example visualization from the simulation environments used in our experiments.

analysis therefore serves as a motivating geometric interpretation and provides qualitative guidance on how base-space regularization affects downstream policy deviation, rather than implying that the surrogate exactly minimizes $D_{KL}(q_u \| \mathcal{N})$.

For Gaussian reference measures, penalizing the quadratic energy together with the volume-change correction suppresses large-norm samples and excessive expansion in typical refinement regimes, which is qualitatively consistent with maintaining a moderate distributional shift.

## C. Environment Details

In this section, we describe our experimental framework and implementation of the proposed method, including benchmark and datasets, task descriptions, evaluation metrics, and training details.

### C.1. Benchmark Details

We use the Datasets for Safe Reinforcement Learning (DSRL) benchmark suite (Liu et al., 2023a) to train and evaluate our method as well as all baselines. DSRL provides 38 offline datasets spanning multiple safe RL environments (Safety-Gymnasium, Bullet-Safety-Gym, and Safe MetaDrive) with varying difficulty levels. These datasets follow a D4RL-style (Fu et al., 2020) API and include detailed cost signals in addition to reward returns. For the baselines, we adopt the authors' official implementations and default hyperparameters when available (especially for FISOR and LSPC). For other methods (BCQL / BCQ-Lag, CPQ, CDT), we use the OSRL framework's implementations and settings to ensure fair comparison.

### C.2. Task Descriptions

We evaluate GSCO on three standard safe offline RL benchmark suites. Figure 6 shows representative task visualizations.

**Safety-Gymnasium.** Safety-Gymnasium (Ji et al., 2023) is a MuJoCo-based safe RL benchmark with continuous-control agents such as Car, Ant, HalfCheetah, and Swimmer. We use tasks including Goal, Button, Push, Circle, and Velocity, which combine locomotion or navigation objectives with safety constraints such as obstacle avoidance, collision costs, and velocity limits.

**Bullet-Safety-Gym.** Bullet-Safety-Gym (Gronauer, 2022) provides PyBullet-based continuous-control safety tasks with agents such as Ball, Car, Drone, and Ant. The evaluated tasks include Run and Circle variants, where agents must complete locomotion objectives while avoiding unsafe contacts or boundary violations under different physics dynamics.

**Safe MetaDrive.** Safe MetaDrive (Li et al., 2022) evaluates safe driving policies in procedurally generated road environments. The tasks require continuous steering and acceleration control while respecting driving-related safety constraints, including collision avoidance, lane keeping, and road-boundary violations. Compared with the other suites, these tasks introduce more structured driving scenarios and a stronger reward–safety trade-off.

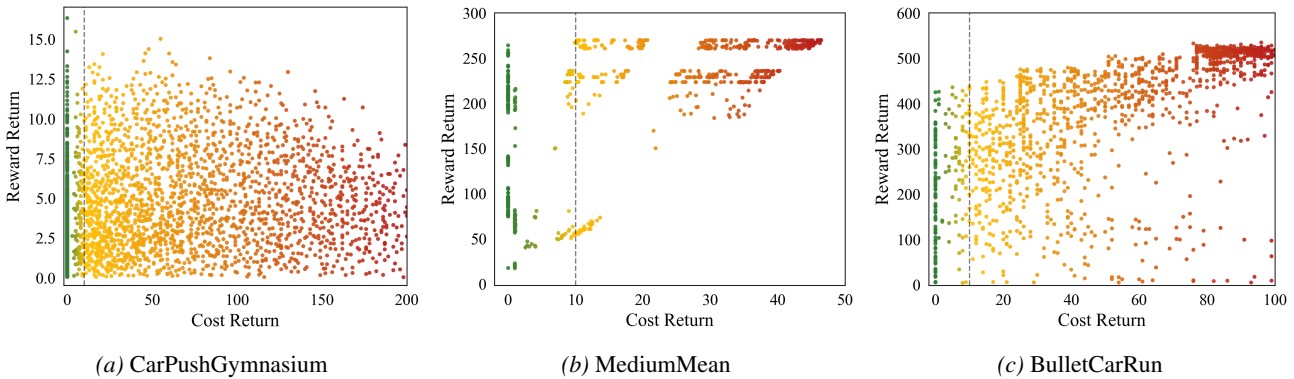

*(a)* CarPushGymnasium           *(b)* MediumMean           *(c)* BulletCarRun

*Figure 7.* Example visualization of the dataset used in our experiments.

## C.3. Dataset Visualization

We further present the distribution of offline trajectories in the cost–return space across three representative environments, as shown in Figure 7. In the `CarPush` task from Safety-Gymnasium, the reward distribution is narrow and low, while the cost spans a wide range. This results in a weak correlation between reward and safety: most trajectories incur high costs even when achieving only modest returns, making strict constraint satisfaction particularly challenging. In the `MediumMean` task from Safe MetaDrive, the reward exhibits distinct discrete bands, each associated with a specific cost level. This reflects mode-switching behaviors and a strong reward–cost coupling; although feasible trajectories exist, achieving high reward under tight cost limits requires careful selection among these behavioral clusters. The `CarRun` task from Bullet Safety Gym demonstrates a smoother trade-off frontier, where reward gradually increases with cost, forming a continuous and diverse distribution. While safe, high-reward trajectories remain sparse, the presence of mid-reward, intermediate-cost episodes renders this dataset more amenable to constrained policy optimization compared to the other two.

## C.4. Evaluation Metrics

We evaluate all methods using *normalized reward return* and *normalized cost return*, following standard practice in offline RL benchmarks such as D4RL (Fu et al., 2020) and recent safe RL works (e.g., CDT (Liu et al., 2023b), LSPC (Koirala et al., 2024), FISOR (Zheng et al., 2024)). The normalized reward is defined as

$$R_{\text{norm}} = \frac{R_\pi - r_{\min}(\mathcal{T})}{r_{\max}(\mathcal{T}) - r_{\min}(\mathcal{T})}, \tag{61}$$

where $R_\pi$ is the return of policy $\pi$, and $r_{\max}(\mathcal{T})$, $r_{\min}(\mathcal{T})$ are the maximum and minimum returns in the dataset.

The normalized cost is computed as

$$C_{\text{norm}} = \frac{C_\pi}{\kappa + \epsilon}, \tag{62}$$

where $C_\pi$ is the total cost, $\kappa$ is the cost limit (set to 10 for all tasks), and $\epsilon$ is a small constant for numerical stability.

## D. Training Details

For all baseline methods, we adopt their default hyperparameter configurations. To ensure a fair comparison across all methods, we set the rollout length for each task to match the maximum number of allowed interaction steps. The cost limit for the baselines is set to 10 for all tasks. The common key hyperparameters used for our method and baselines are shown in Table 7. Table 8 lists other key hyperparameters used for GSCO. We apply the same configuration across all tasks and environments without per-task tuning. The shared geometric refiner is implemented as a single conditional affine coupling layer with a lightweight MLP.

The pseudocode for GSCO is provided in Algorithm 1. All experiments were conducted on eight NVIDIA RTX 6000 Ada Generation GPUs, each with 48 GB of memory. Each experiment is run with 3 random seeds, and results are averaged over 10 evaluation episodes per seed.

*Table 7.* Model Configuration Parameters

| Parameter | CPQ | BCQ-L | CDT | LSPC | FISOR | GSCO |
|---|---|---|---|---|---|---|
| *Common Settings:* | | | | | | |
| Training steps | | | $1 \times 10^6$ | | | |
| Batch size | | | 512 | | | |
| Discount factor | | | 0.99 | | | |
| Activate function | | | ReLu | | | |
| *Algorithm-Specific Settings:* | | | | | | |
| Hidden layer size | 256 | 256 | 256 | 256 | 256 | 256 |
| Soft update rate ($\tau$) | 0.005 | 0.005 | 0.005 | 0.005 | 0.001 | 0.001 |
| Cost limit | 10 | 10 | 10 | – | – | – |
| *Learning Rates ($\times 10^{-3}$):* | | | | | | |
| Actor learning rate | 1.0 | 1.0 | 0.1 | 0.3 | 0.3 | 0.3 |
| Critic learning rate | 1.0 | 1.0 | 0.1 | 0.3 | 0.3 | 0.3 |

*Table 8.* Hyperparameters of GSCO.

| Parameter | Value |
|---|---|
| Expectile $\tau$ | 0.9 |
| Asymmetric L2 loss coeff | 0.9 |
| Target temperature | 3 |
| Value temperature | 5 |
| Advantage weight clip (reward) | $(-\infty, \ 100]$ |
| Advantage weight clip (cost) | $(-\infty, \ 150]$ |
| Refine steps $T$ | 3 |
| Refiner loss weight $\lambda_r, \lambda_h, \lambda_{sh}$ | 1,1,0.5 |

### D.1. Computational Cost

**Architectural simplicity of the flow prior.** The normalizing-flow prior in GSCO is intentionally lightweight. We use a RealNVP-style coupling architecture with affine transformations, whose Jacobian is triangular; hence the log-determinant can be computed in linear time in the latent dimension, without matrix inversion, Hessian computation, inner optimization, or fixed-point iterations. Forward and inverse mappings share the same coupling layers and require only standard first-order neural-network evaluations. Combined with a moderate latent dimension and a small number of coupling layers, this design keeps the flow prior numerically stable and computationally practical while still providing exact likelihoods and invertible latent transformations.

**Comparison with representative baselines.** Table 9 compares GSCO with representative baselines under the same hardware and evaluation setup. GSCO has higher per-step training time than FISOR and LSPC, mainly due to its additional base-space refinement and flow evaluation, but it remains faster than CDT. Its memory footprint is comparable to the strongest generative baselines, and its inference latency remains within a practical range: lower than FISOR under sequential single-action generation, though higher than the lightweight LSPC policy. Overall, GSCO introduces a moderate computational cost for explicit geometric refinement, while remaining practical for offline training and deployment.

## E. Additional Experiments

**Effect of refiner loss weights.** Figure 8 investigates how the relative weights assigned to the three refiners (R, H, SH) affect performance. Overall, GSCO is quite robust: within a broad range of loss weights, the reward and cost curves remain stable without sudden degradation. When the safety refiner H is severely under-weighted (left part of the curves), the policy

*Table 9.* Compute comparison of GSCO and representative baselines under the same hardware/setup. Training time is wall-clock time per optimization step. Inference latency is reported per action; for methods measured with sequential single-action generation, the per-action value is obtained by dividing the total time of a fixed batch by the batch size.

| Method | Train time / step (s) | Peak memory (GB) | Infer. latency / action (ms) |
|---|---|---|---|
| FISOR | 0.022 | 1.703 | 2.664 |
| LSPC | 0.036 | 2.976 | 0.375 |
| CDT | 0.092 | 1.954 | 1.270 |
| GSCO (ours) | 0.067 | 2.659 | 1.714 |

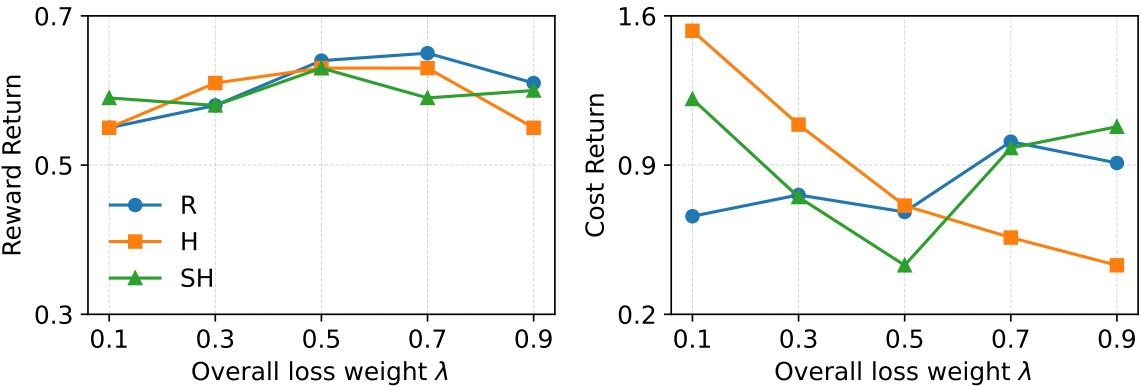

*Figure 8.* Effect of refiner loss weights on GSCO performance: varying the relative weights of the reward (R), safety (H), and shared (SH) refiners yields a robust response and enables a smooth trade-off between reward return and cost.

becomes noticeably less safe, confirming that H is the main driver toward low-cost regions. As the weight of H increases, the policy consistently moves to safer operating points. In contrast, putting more emphasis on the reward refiner R tends to increase the reward return, but also leads to higher cost, which is consistent with its role of exploiting high-return directions near the constraint boundary. The shared refiner SH behaves like a regularizer: when its weight is too small, the policy becomes less coordinated and slightly more unstable; when its weight is too large, over-regularization harms both reward and safety. The best performance is obtained for intermediate SH weights, where it can effectively absorb residual interactions between R and H while keeping the refinement close to the flow prior. These trends show that (i) GSCO's performance is not overly sensitive to the exact choice of refiner weights, and (ii) by tuning the relative weights of R, H, and SH, practitioners can smoothly control the reward–cost trade-off without changing the underlying critics or flow model.

**Decoder freezing ablation.** Freezing the decoder is a core modeling choice in our method: the theoretical coupling between the latent prior and the refiner—and the resulting bounds on action and policy shift—critically rely on the decoder remaining fixed. Allowing the decoder to change would break this coupling and make both the analysis and the interpretation of the refinement steps much less clear. To quantify how much performance is potentially sacrificed by this restriction, we compare our default "frozen decoder" training with an alternative scheme where the refiner and decoder are updated in alternating phases. The result is shown in Table 10a. On the simpler task `CarRun`, the two variants achieve very similar performance: with a frozen decoder, we obtain a reward of 0.87 at zero cost, while alternating updates yield a reward of 0.84, also at zero cost. On the more challenging `AntCircle` task, alternating updates increase the reward from 0.45 to 0.69, but at the price of a higher cost (from 0.25 to 0.56). Thus, while partially unfreezing the decoder can improve returns on complex tasks, it does so by relaxing safety, whereas the frozen-decoder variant preserves our theoretical guarantees and achieves tighter cost control.

**Ablation on refiner optimization strategy.** We further investigate whether the three-refiner architecture is really necessary, or whether one can obtain

*Table 10.* Decoder and refiner ablations.

*(a)* Frozen vs. alternating decoder

| Task | Reward | Cost |
|---|---|---|
| CarRun (frozen) | 0.87 | 0.00 |
| CarRun (alter.) | 0.84 | 0.00 |
| AntCircle (frozen) | 0.45 | 0.25 |
| AntCircle (alter.) | 0.69 | 0.56 |

*(b)* Refiner optimization strategy on `AntCircle`

| Refiner Deisgn | Reward | Cost |
|---|---|---|
| Decoupled 3-refiners | 0.45 | 0.25 |
| Single unified refiner | 0.07 | 0.00 |
| Averaged 3-refiners | 0.51 | 0.45 |

similar behavior by changing only the optimization scheme while keeping the same total loss. On `AntCircle`, we fix the loss weights $(\lambda_r, \lambda_h, \lambda_{sh})$ and compare our default design—three decoupled refiners (H, R, SH) optimized sequentially—with two alternatives (Table 10b): (i) a single unified refiner, which directly optimizes the sum of the three refiner losses, and (ii) an averaged 3-refiner update, where we still learn three refiners but average their latent updates before applying a single step to the base code.

The results show that the three-refiner design is crucial for obtaining a good reward–cost trade-off. The unified refiner collapses to an overly conservative solution (reward 0.07, cost 0.00): because a single set of parameters must simultaneously satisfy safety, reward, and regularization objectives, the gradients from these components frequently conflict, and the optimizer converges to a compromise that prioritizes low cost but fails to exploit high-return directions. By contrast, the averaged-update variant achieves high reward (0.51) but with much higher cost (0.45): averaging the three latent updates at a single point mixes conflicting safety and reward gradients, partially canceling the safety correction and diluting the shared refiner's regularization, which leads to high-return but unsafe solutions. Our sequential H→R→SH updates (0.45 reward, 0.25 cost) strike a substantially better balance that cannot be mimicked by a single averaged step. Overall, these results indicate that separating safety, reward, and shared refiners—each with its own parameters and update direction—is more effective than collapsing them into a single refiner or naively averaging their gradients.

---

**Algorithm 1** GSCO Training (Two-Stage)

---

**Require:** Offline dataset $\mathcal{D}$
1: Init critics $(Q_r, V_r), (Q_h, V_h)$; flow $p_\phi$; decoder $\pi_\theta$; refiners $\{R_s, R_r, R_{\text{sh}}\}$

2: **Stage 1: Critic and flow pretraining**
3: **while** not converged$_{\text{base}}$ **do**
4:     Sample minibatch $(s, a, r, c, s') \sim \mathcal{D}$, draw $z_q \sim q_\psi(z \mid s, a)$

5:     **// Critic updates**
6:     Update safety critics $(Q_h, V_h)$ by HJ-style backup                           ▷ Eq. 31, Eq. 30
7:     Update reward/value critics $(Q_r, V_r)$ by TD / advantage targets     ▷ Eq. 16, Eq. 15

8:     **// Flow prior and decoder**
9:     Compute weighted ELBO, density-shaping, and entropy terms           ▷ Eqs. (4)–(6)
10:    Update $p_\phi, q_\psi$ and $\pi_\theta$ using flow objective $\mathcal{L}_{\text{flow}}$               ▷ Eq. (7)
11: **end while**

12: **Stage 2: Latent refiner training (freeze base model)**
13: Freeze $p_\phi, q_\psi$, and $\pi_\theta$; use target critics for refiner losses.
14: **while** not converged$_{\text{ref}}$ **do**
15:    Sample minibatch $(s, a, r, c, s') \sim \mathcal{D}$ and draw $u_0 \sim \mathcal{N}(0, I)$.
16:    Apply sequential base-space refinement:

$$u_h = u_0 + f_h(s, u_0), \quad u_r = u_h + f_r(s, u_h), \quad u_T = g_\eta(u_r|s).$$

17:    Decode $\tilde{a} = \pi_\theta(s, T_\phi(u_T|s))$.
18:    Update the safety refiner $f_h$ using $\mathcal{L}_h$                               ▷ Eq. (12)
19:    Update the reward refiner $f_r$ using $\mathcal{L}_r$                               ▷ Eq. (11)
20:    Update the shared geometric refiner $g_\eta$ using $\mathcal{L}_{\text{reg}}$            ▷ Eq. (13)
21: **end while**

---

