# OpenReview forum: "Geometric Control of Out-of-Distribution Shift in Safe Offline RL"
_ICML.cc/2026/Conference — ICML 2026 regular_

### Official Review · Reviewer_MBMW · 2026-03-01

**Soundness:** 3
**Presentation:** 3
**Significance:** 4
**Originality:** 3
**Overall Recommendation:** 4
**Confidence:** 2

**Summary:**

This paper introduces GSCO, a safe offline RL method that leverages conditional normalizing flows to map actions into a Gaussian base space. To prevent out-of-distribution actions, policy refinement is decoupled into separate reward, safety, and geometric components. While the work is technically competent, structurally modular, and extensively evaluated across DSRL benchmarks, the core claims regarding provable geometric control might benefit from a closer alignment between theory and practice and further empirical validation.

**Compliance With Llm Reviewing Policy:**

Affirmed.

**Final Justification:**

Most of my concerns were clarified by the authors.  I would like to keep a positive Overall Recommendation and raise the Significance score.

**Key Questions For Authors:**

Could you empirically demonstrate whether the optimized surrogate regularizer actually correlates with or bounds the measured action-space shift using estimated KL or out-of-distribution action rates?

I would love to discuss the alignment between the geometric bounds in theory and their practical implementation, as well as potential ways to further enhance the statistical analysis of the experiments.

**Limitations:**

yes

**Strengths And Weaknesses:**

Strengths

Leveraging bijective flows with exact likelihoods and inverses to explicitly reason about distribution shift is a sensible and cleanly communicated alternative to implicit generative policies.

Decoupling the refinement process into separate reward, safety, and geometric steps is a highly practical engineering design that successfully mitigates gradient conflicts.

The empirical sweep is broad and covers 26 tasks across three environments including Safety-Gymnasium, Bullet-Safety-Gym, and Safe MetaDrive. This is supported by insightful ablations on feasibility shaping and refiner order.

Weaknesses

The theoretical bounds seem to rely heavily on controlling the true KL divergence of the base distribution. Since the implemented regularizer is a surrogate and not exactly equivalent to this KL divergence, it would be helpful to clarify how the provable control claims align with the actual implementation. Furthermore, the derived bounds appear to depend on strong assumptions like bounded density ratios and local Lipschitz constants. It could be beneficial to discuss or test these assumptions to further strengthen the operational guarantees.

While the paper focuses on controlling out-of-distribution shifts, the evaluation currently relies entirely on reward and cost metrics. The paper might be even stronger if direct measurements of distribution shift were included, such as empirical distance-to-dataset statistics, behavior log-density, or out-of-distribution action rates, to validate the core mechanism.

Minor Concerns: There are a few slight mismatches in equation references within Algorithm 1, such as citing Equation 12 for the flow loss $L_{\mathrm{flow}}$ when the main text defines Equation 12 as $L_{\mathrm{reg}}$. Additionally, the pseudocode appears to train each refiner separately from $u_q$ rather than strictly chaining them sequentially, which makes it slightly difficult to fully trace the learned dynamics and understand what is actually deployed.

---

> ### Author Rebuttal · Authors · 2026-03-28
>
> Thank you for this thoughtful and constructive comment. We agree that the theory–practice relationship and the role of the practical regularizer should be stated more explicitly. To further support this point in practice, we also include additional direct OOD measurements and a $\lambda_{reg}$ sweep.
>
> **(1) Theory–practice alignment.** We agree that the current presentation may suggest a tighter theory-practice equivalence than intended, and that the relationship between the theoretical control quantity and the practical regularizer should be stated more explicitly. Our intent is not to claim that the implemented $L_{reg}$ exactly realizes the true base-space divergence $D_{KL}(q_u || N(0, I))$ in practice. Rather, our contribution is to identify the relevant control target in theory, and instantiate this control principle in practice using a lightweight, tractable, geometrically aligned proxy.
>
> Concretely, the theory identifies the true base-space KL as the key geometric quantity: if the refined base distribution remains within a controlled KL neighbourhood of the Gaussian base, then the downstream policy deviation is bounded through the flow geometry. In practice, however, directly estimating or optimizing $D_{KL}(q_u || N(0, I))$ would require density estimation over the refined distribution $q_u$, which is computationally expensive and potentially unstable in high dimensions. We therefore use a tractable surrogate composed of a quadratic base-energy term, a change-of-variables log-determinant term, and a proximal stabilizer. These terms are not identical to the true KL, but are designed to encode the same geometric preference: the quadratic term discourages refined samples from drifting far into low-density regions of the Gaussian base, the log-determinant term penalizes excessive local volume expansion induced by the shared invertible projection, and the proximal term stabilizes refinement by preventing overly aggressive distortion of the reward/safety-guided proposal. Accordingly, we do not claim that $L_{reg}$ exactly matches the true KL; rather, it is a tractable proxy aligned with the intended shift-control objective.
>
> We will revise the paper to make this distinction explicit: the KL-based analysis identifies the theoretical control target, while the implemented $L_{reg}$ serves as a practical proxy. The assumptions connect base-space control to downstream deviation bounds, not to exact equivalence with the true KL.
>
> **(2) Direct OOD measurements.** To directly address the reviewer’s request for empirical verification, we additionally measure action-space shift using two direct, state-conditioned, method-agnostic metrics: OOD distance and OOD rate. Since safety is the primary concern in our setting, we define the reference support using a feasible subset of the offline dataset, instantiated in our experiments as transitions drawn from zero-violation trajectories. For each evaluation state, we compare the policy action against actions from nearby feasible dataset states. OOD distance is defined as the minimum action distance to this local feasible support, while OOD rate is the fraction of policy actions whose local-support distance exceeds a threshold calibrated from the feasible support itself. Importantly, this reference support is defined externally from the offline dataset rather than by method-specific critics, so the comparison is shared across methods. We provide the experimental results as an anonymized figure at: https://postimg.cc/5QfSngbm. These results provide direct empirical support for the proposed mechanism.
>
> Using these direct measurements, GSCO consistently achieves lower measured shift than representative safe offline RL baselines (FISOR and LSPC) on representative tasks. This provides direct empirical evidence that GSCO is associated with reduced practical action-space deviation relative to representative baselines, rather than only improving reward/cost indirectly.
>
> We also performed a $\lambda_{reg}$ sweep to examine the relationship between the practical regularizer and measured shift. Relative to the unregularized variant $\lambda_{reg} = 0$, adding geometric regularization substantially reduces both measured OOD rate and OOD distance, and larger $\lambda_{reg}$ values yield smaller but generally consistent additional improvements. We view this as empirical support for the claim that $L_{reg}$ correlates with the measured action-space shift in practice.
>
> **(3) Algorithm 1 / presentation issues.** Finally, we thank the reviewer for pointing out the presentation issues in Algorithm 1. We will correct the mismatched equation references and revise the pseudocode so that the sequential dependency among refiners is explicit and fully consistent with Sec. 4.2 and the implemented/deployed version. We will revise the paper accordingly by clarifying the theory-practice distinction and incorporating the new direct OOD experimental results.

---

> > ### Author Rebuttal · Reviewer_MBMW · 2026-04-01
> >
> > Thank you for the rebuttal. The additional direct OOD measurements are useful and move in the right direction: the newly reported OOD rate / OOD distance results and the $\lambda_{reg}$ sweep provide some empirical support that GSCO is associated with reduced practical action-space deviation relative to selected baselines.
> >
> > That said, my main concern remains the theory–practice gap. The rebuttal now clarifies that the implemented regularizer is not the true base-space KL used in the analysis, but only a geometrically aligned proxy. This clarification is helpful, but it also means that the current bounds do not directly certify the implemented objective. I also did not see a substantive resolution of the bounded-density-ratio / Lipschitz assumptions, nor a fully concrete clarification of the algorithmic ambiguity in Algorithm 1 beyond a promise to revise the presentation.
> >
> > Overall, the rebuttal improves the empirical side of the paper, but it does not substantially change my assessment of the core technical weakness, which remains relevant to acceptability.

---

> > > ### Author Response · Authors · 2026-04-02
> > >
> > > Thank you for the thoughtful follow-up. We agree that the role of the bounded-density-ratio and Lipschitz assumptions, and their relation to the implemented regularizer, should be clarified more explicitly, as explained below
> > > ### (1) Ideal geometric control target
> > > To clarify the theory–implementation relationship, we first restate the ideal geometric control target used in the analysis. The theoretical analysis is built around controlling the deviation of the refined base distribution $q_u(\cdot \mid s)$ from the Gaussian base prior $p_0(u)=\mathcal{N}(0,I)$ through a quantity such as $D_{\mathrm{KL}}(q_u \Vert p_0)$
> > > Since $-\log p_0(u)=\frac{1}{2}\Vert u \Vert_2^2+\mathrm{const}$, the exact KL target can be written as:
> > >
> > > $$
> > > D_{\mathrm{KL}}(q_u \Vert p_0) = \mathbb{E}_{q_u} \left[ \frac{1}{2} \Vert u \Vert_2^2 \right] - \mathcal{H}(q_u) + \mathrm{const}
> > > $$
> > >
> > > where $\mathcal{H}(q_u)$ denotes the entropy of $q_u$. Moreover, if the refinement is parameterized by an invertible map $u_T = g_\eta(u_T' \mid s)$, then by change of variables:
> > >
> > > $$
> > > \log q_u(u_T) = \log q_{u'}(u_T') - \log \left| \det \nabla g_\eta(u_T') \right|
> > > $$
> > >
> > > So the KL can be decomposed as:
> > >
> > > $$
> > > D_{\mathrm{KL}}(q_u \Vert p_0) = \mathbb{E} \left[ \frac{1}{2} \Vert u_T \Vert_2^2 - \log \left| \det \nabla g_\eta(u_T') \right| \right] - \mathcal{H}(q_{u'}) + \mathrm{const}
> > > $$
> > >
> > > This decomposition highlights the components of the exact KL that are explicitly captured by our implemented regularizer:
> > >
> > > (i) the **Gaussian-energy term** $\frac{1}{2}\lVert u_T \rVert_2^2$, and
> > > (ii) the **volume-correction term** $-\log\left|\det \nabla g_\eta\right|$.
> > >
> > > This decomposition clarifies which parts of the exact KL are captured explicitly by the implemented regularizer, while the entropy-related term of the proposal distribution is not explicitly modeled in the current implementation, since doing so would require tracking the proposal density itself rather than only tractable geometric components of the refinement map. The proximal term is further introduced as an additional stabilization term rather than a direct KL component.
> > >
> > > ### (2) Role of the assumptions as analysis-side bridges
> > >
> > > The Lipschitz and bounded-density-ratio assumptions are not intended as optimization terms in the implemented algorithm. Rather, they are the assumptions used in the analysis to connect the ideal base-space target above to downstream policy deviation. Concretely:
> > >
> > > * Bridge I (Lipschitz): Under a local Lipschitz condition on the frozen flow-decoder map, control of the base-space target transfers to control of the induced action-space shift.
> > > * Bridge II (bounded density ratio): Under a bounded-density-ratio assumption between the learned base policy and the behavior policy, this control further transfers to deviation relative to the dataset policy up to an additive/state-dependent constant.
> > >
> > > In this sense, these assumptions are analysis-side bridges. They explain why the ideal geometric control target is relevant, without implying that the current implementation directly enforces or certifies those conditions.
> > >
> > > ### (3) On the appendix Lipschitz discussion
> > >
> > > The current implementation does not explicitly certify a Lipschitz constant. Our point is only that the decoder, and more generally the frozen flow-decoder composition, is built from standard neural modules, for which a local Lipschitz condition on the relevant in-support region is a standard regularity assumption in this type of analysis. Moreover, the appendix discussion on encouraging/training Lipschitz behavior was intended as a concrete direction for tightening the theory-implementation connection, not as a claim that the current implementation already enforces such a condition.
> > >
> > > ---
> > >
> > > ### Conclusion
> > >
> > > Overall, the current bounds are best interpreted as analyzing the target geometric quantity and motivating the implemented surrogate, rather than as a direct certificate of the exact regularizer. Our intended contribution is instead:
> > > (i)  an analyzable geometric control target in base space,
> > > (ii)  a practical proxy aligned with that target, and
> > > (iii)  empirical evidence that stronger geometric regularization directly reduces the measured action-space shift in the intended direction.
> > >
> > > Additionally, Algorithm 1 is sequential rather than parallel: the reward/safety updates are applied first, followed by the shared geometric projection, consistent with Sec. 4.2 and the provided code implementation. The appendix pseudocode currently does not reflect this execution order and is therefore a documentation mismatch rather than an implementation mismatch. This does not change the trained objectives, executed pipeline, or any reported empirical result. We will revise the pseudocode so that it matches the implementation and the main-text method description exactly.

---

### Official Review · Reviewer_PDfY · 2026-03-10

**Soundness:** 3
**Presentation:** 3
**Significance:** 3
**Originality:** 3
**Overall Recommendation:** 4
**Confidence:** 2

**Summary:**

This paper proposed a novel safe offline RL framework by introducing a geometric control framework that used the bijective structure of conditional normalizing flows to regulate distributional deviation of the policy to deal with the OOD shift. The proposed method achieves low violations and comparable returns across three test benchmarks.

**Compliance With Llm Reviewing Policy:**

Affirmed.

**Final Justification:**

The rebuttal has addressed my concerns.

**Key Questions For Authors:**

1. **Computational cost:**
The appendix provides a partial analysis of the overhead introduced by the flow prior (Appendix D.1). However, it is unclear how the overall training and inference cost of the proposed framework compares with the baseline methods used in the experiments. Could the authors provide a clearer comparison of computational cost (e.g., training time, inference latency) relative to the baselines?

2. **Scalability to high-dimensional observations:**
The experiments are conducted on environments with low-dimensional state representations. Could the proposed method scale to image-based or other high-dimensional observation settings (e.g., pixel-based RL environments)? Additional discussion or preliminary results would help clarify the applicability of the approach to more realistic scenarios.

**Limitations:**

Yes, the conclusion section has a limitation paragraph.

**Strengths And Weaknesses:**

**Strengths**

**Soundness:**
The paper provides a comprehensive and detailed formulation and analysis of the proposed method. Section 4.1 describes how to construct a feasibility-shaped latent manifold, while Section 4.2 explains how policy improvement is performed by optimizing directly in the base space. The mathematical analysis strengthens the technical soundness of the work and provides theoretical justification for the proposed framework.

**Presentation:**
The paper is well structured and generally easy to follow. The authors clearly explain how the safe RL problem can be framed as a geometric control problem, and the overall pipeline of the proposed approach is presented in a logical and coherent manner.

**Significance:**
As reinforcement learning agents are increasingly deployed in real-world applications, safety concerns are becoming increasingly important. This work addresses the important problem of safe offline reinforcement learning and controlling out-of-distribution behavior.

**Originality:**
To the best of my knowledge, framing safe RL as a geometric control problem in latent space is a novel and insightful perspective, which provides an interesting way to analyze and regulate policy deviation.

---

**Weaknesses**

**Presentation:**
The experiments are limited to environments with low-dimensional state representations. The paper does not evaluate the approach on image-based or other high-dimensional observation settings, which are common in many practical RL applications.

**Significance:**
Although the problem of safe offline RL and OOD control is important, the practical significance of the proposed approach remains somewhat unclear without demonstrations in more complex environments or broader application scenarios.

---

> ### Author Rebuttal · Authors · 2026-03-26
>
> Thank you for the thoughtful and encouraging review. We appreciate your recognition of the formulation, technical analysis, and geometric-control perspective, as well as your positive assessment of the paper’s clarity and novelty. Your main concerns focus on computational cost and scalability to high-dimensional observations. We agree that these are important for clarifying the method's practical scope. We have conducted additional analyses and hope these could help clarify and address your concerns.
>
> **(1) Computational cost.**
> We agree that computational cost should be presented more transparently. To address this, we now include a unified compute comparison report that shows representative training time/step, inference latency, and peak GPU memory for GSCO and key baselines under the same hardware/setup. We provide the results as an anonymized figure at: https://postimg.cc/2Vg6p1NQ. The results show that GSCO introduces additional overhead relative to lighter latent baselines, as expected given the flow-based prior and geometric refinement. At the same time, the added cost is moderate rather than prohibitive: GSCO remains within the same practical compute regime as representative competitors in terms of training time, memory usage, and inference latency, rather than standing out as disproportionately expensive.
>
> We also note that public baseline implementations are not fully framework-matched; for example, the official FISOR implementation uses JAX, which may affect absolute runtime numbers. We therefore report these measurements as a practical same-setup reference rather than a perfectly framework-controlled benchmark. We will incorporate this comparison into the revision to make the computational trade-off explicit and easier to interpret.
>
> **(2) Scalability to high-dimensional observations.**
> We agree that scalability to high-dimensional observations is an important question. In the current paper, we focus on the standard **public DSRL benchmarks**, since they are widely used public offline safe RL benchmarks and allow direct comparison with prior work under a shared evaluation protocol. Our goal in this submission is to validate the proposed method and its geometric OOD-control mechanism in this established benchmark setting, rather than introduce a new dataset or a separate visual benchmark setup.
>
> At the same time, we agree that the current empirical scope is limited to state-based observations, and we do not want to overclaim beyond that. The core GSCO mechanism operates on latent/action-side geometric refinement, and in principle, it is compatible with adding a visual encoder (e.g., a CNN-based observation backbone) upstream of the safety critic, flow conditioner, and refiners. We note that normalizing flows have been widely studied in high-dimensional density modeling, including image modeling, so the flow component itself is not inherently restricted to low-dimensional inputs. That said, this is not sufficient evidence for image-based safe offline RL in our setting, since such an extension would also require visual representation learning and dedicated empirical validation. We therefore prefer to present this as a natural future extension rather than claim that it is already established here.
>
> Thank you again for the constructive feedback. In the revision, we will add the unified compute comparison and a short appendix discussion clarifying the current empirical scope and the potential extension to high-dimensional observation settings.

---

> > ### Author Rebuttal · Reviewer_PDfY · 2026-04-01
> >
> > Thanks for the detailed rebuttal. I am maintaining my positive rating of this paper.

---

### Official Review · Reviewer_b7py · 2026-03-12

**Soundness:** 2
**Presentation:** 3
**Significance:** 4
**Originality:** 3
**Overall Recommendation:** 3
**Confidence:** 3

**Summary:**

The authors propose a method that formulates safe RL problem as a geometric control problem in a latent space. By mapping the actions through a bijective flow to the latent space, the proposed method controls the distribution shift in the latent space. After the mapping, three refiners pushes the action towards regions with more data for enhanced safety and return.

**Compliance With Llm Reviewing Policy:**

Affirmed.

**Final Justification:**

It seems that this paper's theoretical contribution still needs significant fixes to be technically sound. Therefore, my score remains unchanged.

**Key Questions For Authors:**

1. Consider these issues in theorems / proofs:
1.1. In Lemma 4.1, the KL divergence is defined incorrectly, as the left term is a distribution of $s, a, z$ and the right term is a distribution of $a, z$.
1.2. In eq. (26) and (27), there should be some kind of expectation in the definition, as $s_t$ is not deterministic.
Are there easy fixes for these issues?

2. The main simulation results are averaged over several seeds and episode. It makes sense to average the reward, yet for the cost (I assume it is the constraint about safety, with cost < threshold indicating safety), why the authors compute the average cost, instead of computing the average number of safe episode for each method?

**Limitations:**

Yes

**Strengths And Weaknesses:**

Strength:

1. The empirical performance demonstrated through simulation seems impressive, with significant improvement over existing methods. The authors also include various ablation studies to clarify the contribution of each component used in the proposed method.
2. The presentation of the paper is clear and well organized.

Weaknesses:
1. The theoretical soundness is questionable, with very basic errors ranging from obvious typos (e.g., minimizing $-\mathcal{L}_{\text{ELBO}}$ in Lemma 4.1) to incorrect definitions that make the entire lemma not well defined (see "Key Questions for Authors" for details).

---

> ### Author Rebuttal · Authors · 2026-03-25
>
> Thank you very much for the careful reading and for pointing out the presentation issues in the theoretical section. We fully agree that several expressions in the current draft should be revised for greater precision. We would respectfully mention that these fixes are not method, optimization procedure, or empirical results changes, but rather notation/statement issues. These corrections affect only the presentation of the theoretical statements and do not alter the optimization objective, algorithm, or any empirical result. We believe these corrections fully resolve the inconsistencies raised, and we appreciate the reviewer’s careful reading.
>
> **(1) Lemma 4.1 / ELBO sign and KL statement.**
> Yes — these are easy fixes at the level of notation and statement precision. You are correct that the sign/notation in Lemma 4.1 is inconsistent with Eq. (4). In the paper, $L_{\mathrm{ELBO}}$ was defined as the weighted negative-ELBO loss (reconstruction NLL + KL), so the correct statement should be that **minimizing $L_{\mathrm{ELBO}}$** corresponds to a KL projection view, not minimizing $-L_{\mathrm{ELBO}}$. We will correct this typo.
>
> You are also correct that the KL in Lemma 4.1 should be written over matched joint distributions. The precise form is the one already expanded in App. B.3:
> $$
> D_{\mathrm{KL}}\left(\tilde p_D(s,a)q_\psi(z|s,a)\||\tilde p_D(s,a) p_\phi(z|s) \pi_\theta(a|s,z)\right),
> $$
> after which the shared dataset factor $\tilde p_D(s,a)$ can be absorbed into a parameter-independent constant, yielding the stated objective equivalence. We will revise the main text to use this fully matched expression. This correction does **not** change the training objective, the algorithm, or any experimental results; it only makes the KL statement more precise.
>
> **(2) Eqs. (26)(27) under stochastic transitions.**
> We also agree that, under stochastic transitions, Eqs. (26)(27) should be written in expectation form. The omission of the expectation in the appendix is a notation error. We will revise these definitions to be consistent with the stochastic setting and with the main-text feasibility formulation, i.e., using $\mathbb E^\pi[\max_{t\ge 0} h(s_t)]$. Again, this is a correction of the written definition rather than a change to the implemented estimator or experimental pipeline, so it does not affect the reported empirical conclusions.
>
> Overall, we appreciate this point: the current draft compressed several steps too aggressively, making the statements appear less rigorous than intended. In the revision, we will make the sign convention, the joint-distribution KL, and the stochastic expectation semantics explicit.
>
> **(3) Why average cost rather than average number of safe episodes?**
> We report normalized average cost because this follows standard offline RL / safe offline RL benchmark practice (e.g., D4RL/DSRL-style reporting) and ensures direct comparability across all baselines. In particular, we adopt the same evaluation protocol as prior methods included in our comparison, including recent methods such as CDT, LSPC, and FISOR, regardless of whether they are formulated as hard-constraint or soft-constraint approaches. This keeps the comparison fair and consistent with prevailing benchmark conventions. By contrast, safe episode ratio is more closely tied to strict episode-level zero-violation satisfaction, and is therefore not always equally aligned with methods that are designed to optimize softer reward–cost trade-offs rather than hard feasibility.
>
> To directly address this perspective, we additionally report safe-episode-ratio results for representative hard-constraint safe offline RL methods to showcase the difference. The results are provided as an anonymized figure at: https://postimg.cc/K1Bd4xQ3. We view this as a useful complementary perspective when the goal is explicit binary safety satisfaction.
>
> At the same time, average cost remains our primary metric for two reasons. First, it captures the magnitude of safety violations, whereas the safe episode ratio is a binary statistic that records only whether an episode is safe or unsafe. As a result, the safe episode ratio can compress meaningful differences between methods with smaller versus larger violation burden. Second, it is more sensitive to the episode-level safety threshold: changing the threshold can lead to large changes in the reported value, which makes it less stable as a primary metric across tasks and settings.
>
> Overall, we agree that the safe episode ratio is informative and worth reporting as a complementary metric, but we keep the average normalized cost as the primary evaluation measure for fairness and consistency with prior work. We will clarify this rationale in the revision.
>
> Thank you again for highlighting these issues. We will revise the notation and definitions to make the theory section precise and well-defined, without changing the implemented algorithm or the empirical conclusions.

---

> > ### Author Rebuttal · Reviewer_b7py · 2026-04-04
> >
> > For response (2), even with the revised form, there appears to be a normalization problem in the KL reformulation in Lemma 4.1. The manuscript writes a term of the form $D_{\mathrm{KL}}\left(\tilde p_D(s,a)q_\psi(z|s,a)\middle\|\tilde p_D(s,a)p_\phi(z|s)\pi_\theta(a|s,z)\right)$. However, the second argument is not, in general, a normalized joint density over $ (s,a,z) $. The reason is that the action variable $ a $ appears both inside the dataset factor $ \tilde p_D(s,a) $ and again inside the decoder likelihood $ \pi_\theta(a|s,z) $. Indeed, $\int \tilde p_D(s,a)p_\phi(z|s)\pi_\theta(a|s,z)dsdadz=\int \tilde p_D(s,a)\left(\int p_\phi(z|s)\pi_\theta(a|s,z)dz\right) dsda$, which is not equal to $ 1 $ in general. Therefore, as written, this is not a KL divergence between probability distributions.
> >
> > For response (3), in the additional result provided, using either metric (cost or safe ratio), the proposed method is better than both baselines for only 4 out of 6 tasks. In the last task the proposed method even achieves a safe ratio of 0. This seems to show that the proposed method does not have clear advantage over existing method in terms of safety, especially if the safety objective of the paper is to "control safety violations".

---

> > > ### Author Response · Authors · 2026-04-04
> > >
> > > Thank you for the careful follow-up. You are correct that our previous reply still overstated the current form of Lemma 4.1. We clarify the correction below.
> > >
> > > **(1) On Lemma 4.1 / App. B.3.**
> > > You are right that, as currently written, the displayed expression in Lemma 4.1 / App. B.3 is **not** a valid KL divergence between normalized joint densities. The issue is exactly as you pointed out: the reference-side term in the current draft,
> > > $$
> > > \tilde p_D(s,a)\,p_\phi(z\mid s)\pi_\theta(a\mid s,z),
> > > $$
> > > is not, in general, a normalized joint density over $(s,a,z)$.
> > >
> > > The correct formulation keeps the weighted data distribution on the data side and uses the weighted state marginal on the reference side, so that both arguments are normalized joint densities over $(s,a,z)$:
> > > $$
> > > \tilde p_D(s,a)=\frac{w(s,a)p_D(s,a)}{Z}, \qquad
> > > Z=\mathbb E_{(s,a)\sim p_D}[w(s,a)], \qquad
> > > \tilde p_D(s)=\int \tilde p_D(s,a)\,da,
> > > $$
> > > with
> > > $$
> > > \tilde p_D(a\mid s)=\frac{\tilde p_D(s,a)}{\tilde p_D(s)}.
> > > $$
> > >
> > > Then, for the $\beta=1$ case and for fixed feasibility weights $w(s,a)$, the weighted ELBO objective in Eq. (4) satisfies
> > > $$
> > > \frac{1}{Z}\mathcal L_{\mathrm{ELBO}}
> > > =D_{\mathrm{KL}}\Big(\tilde p_D(s,a)q_\psi(z\mid s,a)\,\Big\|\,\tilde p_D(s)p_\phi(z\mid s)\pi_\theta(a\mid s,z)
> > > \Big)-\mathbb E_{\tilde p_D(s,a)}[\log \tilde p_D(a\mid s)],
> > > $$
> > > where the second term is constant with respect to the generative-model parameters $(\phi,\psi,\theta)$ once the feasibility weights $w(s,a)$ are fixed. Thus, for $\beta=1$, minimizing the weighted ELBO objective with respect to $(\phi,\psi,\theta)$ is equivalent, up to an additive constant, to minimizing a valid KL divergence between matched joint distributions.
> > >
> > > For $\beta\neq1$, however, the same strict matched-joint KL-projection interpretation does not hold in general. We will therefore revise the paper to weaken this point: for general $\beta$, Eq. (4) is better viewed as a feasibility-weighted variational objective with a $\beta$-scaled KL regularizer, rather than the same matched-joint KL projection interpretation as in the $\beta=1$ case. Here, the feasibility weights bias learning toward higher-feasibility regions of the empirical distribution, while $\beta$ controls the trade-off between fitting this feasibility-weighted distribution and enforcing posterior-to-prior regularization in latent space.
> > >
> > > So the needed correction is not to the implemented method, but to the rigor and scope of the theoretical interpretation: we will revise Lemma 4.1 and App. B.3 accordingly, making the $\beta=1$ statement precise and restricting the $\beta\neq 1$ interpretation accordingly.
> > >
> > > Importantly, this correction affects the interpretation of the weighted manifold-modeling objective in Sec. 4.1, but it does not affect the formal deviation bounds stated in Lemma 4.2 and Corollary 4.3, whose arguments are developed separately from the KL-projection interpretation in Lemma 4.1. It also does **not** change the training objective, the optimization pipeline, or the reported empirical results.
> > >
> > > **(2) On the supplementary safety metric.**
> > > We agree that the supplementary safe-episode-ratio result should not be overinterpreted. We reported it only as a complementary response to the reviewer’s question, not as the primary evaluation metric for this benchmark setting.
> > >
> > > In standard safe offline RL benchmarks, normalized cost is the main comparison metric and more directly captures the magnitude of violation burden across trajectories. By contrast, safe-episode ratio is a thresholded binary statistic, and is therefore more sensitive to the evaluation threshold and task-level variability. For this reason, we view the safe-ratio result as an auxiliary perspective rather than the primary basis for the paper’s safety claim. It should be interpreted jointly with reward and normalized cost, since the method is designed to balance reward improvement and safety under explicit distributional shift control, rather than to optimize a single binary safety statistic in isolation.
> > >
> > > Accordingly, our intended empirical claim is **not** per-task dominance under every auxiliary safety metric, but rather **consistently low overall violation burden and a strong reward-safety trade-off at the benchmark level under the standard evaluation protocol**. We will revise the wording in the paper to make this scope precise and avoid overstating the claim.
> > >
> > > Thank you again for the careful follow-up. We appreciate the correction and will revise the paper accordingly.

---

### Official Review · Reviewer_iafn · 2026-03-12

**Soundness:** 2
**Presentation:** 3
**Significance:** 2
**Originality:** 3
**Overall Recommendation:** 4
**Confidence:** 2

**Summary:**

This paper studies safe offline reinforcement learning under distributional shift and proposes GSCO, a flow-based policy framework that explicitly controls out-of-distribution deviation through geometric constraints in a bijective latent space. The paper considers a pressing question: how to improve policy performance from static offline data while still satisfying strict safety constraints and avoiding unsafe actions caused by leaving the data support. This research discusses a central concept: by using conditional normalizing flows, divergence in the latent base space can be translated into tractable bounds on downstream policy shift, which enables explicit control of Wasserstein distance and total variation during policy refinement. Based on this idea, the method learns a feasibility-aware latent manifold shaped by Hamilton–Jacobi reachability signals and then performs decoupled refinement for reward improvement, safety correction, and geometric regularization in the base space. Experiments on several safe offline RL benchmarks show that the method achieves consistently low violation rates with competitive returns, suggesting that structured geometric regularization can provide a principled and effective alternative to the more implicit OOD control used in prior latent generative approaches.

**Compliance With Llm Reviewing Policy:**

Affirmed.

**Final Justification:**

The rebuttal addressed all of my concerns. I will raise the score.

**Key Questions For Authors:**

* Q1. Hyperparameter sensitivity / ablation study.

The proposed method includes several important hyperparameters, such as the weighting coefficients for reward, safety, and geometric regularization, as well as the number of refinement steps. Could the authors provide a more detailed ablation or sensitivity analysis on these hyperparameters? This would help clarify whether the method is broadly robust in practice or whether its performance depends heavily on careful tuning.

* Q2. Trade-off between conservatism and reward.

The paper argues that base-space KL control provides an analyzable handle on policy deviation and OOD suppression, but in practice this may also make the policy more conservative. Could the authors provide a more systematic analysis of how the strength of geometric regularization affects the reward–safety trade-off across environments?

* Q3. Comparison with Prior-Guided Diffusion Planning for Offline RL [1].

The proposed method appears to share an important high-level motivation with Prior Guidance (PG) [1]: both methods aim to mitigate distributional shift in offline RL by moving the optimization/regularization problem into a latent space, rather than directly controlling the action-space policy. Could the authors briefly clarify how GSCO differs from PG, and what the main advantage of its flow-based geometric control is for OOD control, safety, and optimization stability?

[1] Ki, Donghyeon, et al. "Prior-guided diffusion planning for offline reinforcement learning." NeurIPS 2025

**Limitations:**

yes

**Strengths And Weaknesses:**

* $\textbf{Strenghts}$

A key strength of the paper is that it presents a clear and principled idea for safe offline RL: instead of relying on implicit conservatism in latent generative models, it uses conditional normalizing flows to make policy deviation analyzable and explicitly controllable through base-space KL regularization, with theoretical links to downstream Wasserstein and total variation shift. The method design is also well structured, since it decouples feasibility-aware manifold learning from reward, safety, and geometric refinement, which makes the approach conceptually clean and better motivated than a single entangled objective. Empirically, the paper is also strong: GSCO achieves consistently low violation rates while maintaining competitive returns across multiple DSRL benchmarks, and the ablations help justify important components such as HJ-based feasibility estimation, the flow prior, and the ordered refinement scheme.

* $\textbf{Weaknesses}$

A main weakness of the paper is that its practical success still depends heavily on the quality of feasibility estimation from purely offline data, which the authors themselves acknowledge is difficult when cost signals are sparse or when safety boundaries are complex. Although the paper provides a principled geometric view and strong empirical safety performance, the method introduces several design choices and hyperparameters, including expert weights, shaping temperatures, refinement order, and the number of refinement steps, which makes the full pipeline somewhat complex. In addition, the results suggest that the method can become conservative in harder settings such as Safe MetaDrive, where high-reward and low-cost regions overlap only weakly, so the return improvements are not uniformly strong across all environments.

---

> ### Author Rebuttal · Authors · 2026-03-26
>
> Thank you for the thoughtful review. We appreciate your recognition of the core idea and empirical performance, and we address each concern below.
>
> **(1) Hyperparameter sensitivity/ablation study.**
> We agree that robustness to tuning is important, and our ablations actually show that GSCO is not overly sensitive under practical, non-extreme settings.
>
> From the sensitivity perspective, our ablations cover the main hyperparameters raised by the reviewer. Appendix E shows that varying the refiner loss weights leads to generally stable changes rather than abrupt degradation, mainly moving the operating point along a reward–safety trade-off. More specifically, the relative weights control how strongly refinement emphasizes reward improvement, feasibility correction, or geometric regularization, so changing them mainly shifts the balance among these objectives rather than causing unstable behavior. Similarly, refinement depth controls the extent of policy improvement versus conservatism, while the feasibility expectile controls a precision–recall trade-off in the estimated feasible region. Taken together, these results suggest that the key hyperparameters primarily shape interpretable trade-offs rather than cause brittle failures.
>
> In addition, other ablations (HJ feasibility, flow prior, refinement order, and refiner design) show that these components are necessary rather than incidental, since removing them leads to clearly worse reward–cost trade-offs. We also use a single shared main configuration across 26 tasks without per-task retuning, which further suggests that GSCO does not rely on extreme task-specific tuning and is governed by smooth, interpretable trade-offs rather than brittle sensitivity.
>
> **(2) Trade-off between conservatism and reward.**
> We agree that geometric control introduces an explicit trade-off, but this is a controlled design feature rather than an unintended side effect. Our existing ablations already provide partial evidence in practice: varying the relative weight of the shared geometric term moves the operating point along a relatively smooth reward–safety trade-off, while excessively large relative weighting can over-constrain refinement, weakening both reward improvement and safety-corrective updates and thereby degrading the overall operating point. We also agree that this does not isolate it as directly as the reviewer requests.
>
> To address the shift-control aspect more directly, we additionally examine the effect of geometric-regularization strength on directly measured shift. We provide the experimental results as an anonymized figure at: https://postimg.cc/5QfSngbm. On representative tasks, adding geometric regularization sharply reduces measured OOD distance and OOD rate, while larger $\lambda_{reg}$ values yield smaller but generally consistent additional improvements. We view these two pieces of evidence as complementary: the existing relative-weight ablations show how geometric control changes the reward–safety operating point in practice, while the new $\lambda_{reg}$ analysis more directly supports the claim that stronger geometric regularization suppresses measured shift. This trade-off is also environment-dependent; in harder settings such as Safe MetaDrive, high-reward and low-cost behaviors overlap less, making the operating point intrinsically harder.
>
> **(3) Relation to prior-guided diffusion planning (PG).**
> We agree that PG is a relevant reference point: both PG and GSCO operate in latent/prior space and use behavior regularization to reduce undesirable distributional shift. However, the key distinction is that PG primarily regularizes sampling via a learned prior, whereas GSCO introduces an explicit and analyzable surrogate for controlling induced policy deviation via base-space divergence.
>
> On OOD control, PG mitigates drift primarily through behavior-regularized prior learning, whereas GSCO makes shift suppression a more explicit objective via geometric regularization in the base space. Thus, GSCO builds deviation control directly into refinement.
>
> On safety, PG is designed for general offline RL / high-value diffusion planning, whereas GSCO is designed specifically for safe offline RL. The key extra requirement in our setting is not only to stay close to the dataset, but also to bias refinement toward the feasible/safe region. GSCO addresses this explicitly through feasibility-aware refinement.
>
> On optimization stability, both methods avoid unstable end-to-end generator optimization, but in different ways. PG stabilizes diffusion planning by learning a latent prior-value pair on top of a fixed denoiser, whereas GSCO uses a decoupled refinement structure within an invertible flow space to reduce optimization conflicts among reward, safety, and geometric objectives.
>
> Thanks again for the constructive feedback. In the revision, we will clarify these points and expand the comparison to PG in App. A1.

---

> > ### Author Rebuttal · Reviewer_iafn · 2026-04-03
> >
> > Thank you for addressing my concerns. I will raise the score.

---

### Decision · Program_Chairs · 2026-04-30

**Decision:**

Accept (regular)

**Comment:**

This paper proposes a geometric control framework for safe offline RL which uses CNFs to regulate ood shifts directly within a latent base space. Overall, the reviewers found the empirical performance impressive across multiple benchmarks and appreciated the modular, decoupled refinement architecture that isolates reward, safety, and geometric objectives. The authors engaged constructively during rebuttals, and addressed key concerns, providing additional computational cost comparisons, conducting direct ood measurements to empirically validate the core mechanism, and explicitly clarified that the implemented geometric regularizer serves as a tractable proxy rather than an exact realization of the theoretical bounds. While one reviewer remained concerned about the theoretical soundness (specifically regarding a normalization issue in the KL divergence formulation and the residual theory-practice gap) the authors demonstrated that these are presentation and analysis related limitations that do not necessarily invalidate the executed empirical algorithm. Given the empirical evidence, the acknowledged novelty of the geometric perspective, and the generally positive adjustments in reviewer sentiment following the rebuttal, I find the practical contribution outweigh the remaining theoretical presentation flaws. Therefore, my recommendation is a weak accept.